# Perioperative Pain Management and Opioid Stewardship: A Practical Guide

**DOI:** 10.3390/healthcare9030333

**Published:** 2021-03-16

**Authors:** Sara J. Hyland, Kara K. Brockhaus, William R. Vincent, Nicole Z. Spence, Michelle M. Lucki, Michael J. Howkins, Robert K. Cleary

**Affiliations:** 1Department of Pharmacy, Grant Medical Center (OhioHealth), Columbus, OH 43215, USA; 2Department of Pharmacy, St. Joseph Mercy Hospital Ann Arbor, Ypsilanti, MI 48197, USA; kara.brockhaus@stjoeshealth.org; 3Department of Pharmacy, Boston Medical Center, Boston, MA 02118, USA; william.vincent@bmc.org; 4Department of Anesthesiology, Boston University School of Medicine, Boston Medical Center, Boston, MA 02118, USA; nicole.spence@bmc.org; 5Department of Orthopedics, Grant Medical Center (OhioHealth), Columbus, OH 43215, USA; michelle.lucki@ohiohealth.com; 6Department of Addiction Medicine, Grant Medical Center (OhioHealth), Columbus, OH 43215, USA; michael.howkins@ohiohealth.com; 7Department of Surgery, St. Joseph Mercy Hospital Ann Arbor, Ypsilanti, MI 48197, USA; robert.cleary@stjoeshealth.org

**Keywords:** pain management, opioid stewardship, perioperative care, postoperative pain, multimodal analgesia, regional anesthesia, preemptive analgesia, perioperative medication management, transitions of care, opioid-related adverse effects

## Abstract

Surgical procedures are key drivers of pain development and opioid utilization globally. Various organizations have generated guidance on postoperative pain management, enhanced recovery strategies, multimodal analgesic and anesthetic techniques, and postoperative opioid prescribing. Still, comprehensive integration of these recommendations into standard practice at the institutional level remains elusive, and persistent postoperative pain and opioid use pose significant societal burdens. The multitude of guidance publications, many different healthcare providers involved in executing them, evolution of surgical technique, and complexities of perioperative care transitions all represent challenges to process improvement. This review seeks to summarize and integrate key recommendations into a “roadmap” for institutional adoption of perioperative analgesic and opioid optimization strategies. We present a brief review of applicable statistics and definitions as impetus for prioritizing both analgesia and opioid exposure in surgical quality improvement. We then review recommended modalities at each phase of perioperative care. We showcase the value of interprofessional collaboration in implementing and sustaining perioperative performance measures related to pain management and analgesic exposure, including those from the patient perspective. Surgery centers across the globe should adopt an integrated, collaborative approach to the twin goals of optimal pain management and opioid stewardship across the care continuum.

## 1. Introduction

Surgery is an indispensable part of healthcare, and over 300 million surgical procedures are performed around the world annually [1]. Despite tremendous benefits to survival and quality of life, surgical procedures frequently result in acute pain, among other risks. Suboptimal postoperative pain management is associated with worsened humanistic and economic outcomes, including the development of chronic pain and opioid dependence [2]. In the U.S., opioid analgesics have been the cornerstone of postoperative pain management, driven by earlier efforts to improve treatment of pain and societal expectations for surgical recovery [3,4,5]. The significant risks and costs associated with opioid overuse are now better understood: opioid-related adverse events frequently potentiate complications in postoperative populations and postsurgical opioid prescribing patterns have contributed to the modern U.S. opioid epidemic [6,7,8,9,10,11]. Postoperative opioid prescribing in the U.S. remains alarmingly high and in stark contrast to that of non-U.S. countries, underscoring the need for more widespread adoption of multimodal analgesia and enhanced recovery strategies by American centers [4,12,13,14].

Perioperative pain management and opioid stewardship are therefore comparable in necessity and interrelated in execution. To this end, many organizations have offered guidance on components of their application. This has included general postoperative pain management [15,16,17], perioperative management of patients on preoperative opioids [18], surgery-specific guidelines [19,20,21,22,23,24], medication-specific recommendations [25,26], conceptual frameworks for opioid stewardship [27,28,29], collaborative postoperative opioid prescribing guidelines [30,31,32], statements on perioperative opioid use [33,34], legal opioid prescribing limits [35], and various quality measures for healthcare institutions [36,37,38]. Despite the multitude of recommendations available, a large proportion of surgical patients report inadequately treated pain and high rates of adverse events, alongside many institutions exhibiting overreliance on opioids and underutilization of multimodal strategies [2,39,40]. This narrative review enhances awareness and adoption of perioperative pain management and opioid stewardship strategies by integrating available guidance into a single “roadmap” for interprofessional stakeholders across the surgical care continuum.

## 2. Statistics and Definitions

### 2.1. The Burdens of Perioperative Opioid Overuse and of Uncontrolled Postoperative Pain

Approximately one out of every ten opioid-exposed postoperative patients will experience at least one opioid-related adverse event (ORAE), conferring significant morbidity and economic burden [7,41]. Many postoperative complications may be appropriately classified as ORAEs, including nausea and vomiting, ileus, urinary retention, delirium, and respiratory depression, underscoring the interrelatedness of perioperative opioid use and surgical outcomes [6,41]. Despite their toxicities, opioids appear to be overprescribed for postoperative pain [42,43,44,45,46,47]. Available data suggest 42–71% of prescribed opioid pills go unused after surgery, with 73% of postoperative orthopedic patients reporting unused opioid pills at one month post-procedure [42,46]. This reservoir of unused prescription opioids in community settings has been identified as a potential contributor to the U.S. opioid epidemic. Over 80% of modern heroin users report nonmedical prescription opioid use prior to heroin initiation, and two-thirds of prescription opioids used for nonmedical purposes are obtained from a friend or relative [11,48,49].

Despite an apparent overreliance on opioids by prescribers, less than half of postoperative patients endorse adequate pain relief, with 75–88% reporting a pain severity of moderate, severe, or extreme [2,15]. Short-term morbidities related to uncontrolled acute postoperative pain span nearly every organ system, including increased risks for thrombotic events, pneumonia, ileus, oliguria, and impaired wound healing. Furthermore, inadequate acute pain control negatively impacts long-term functional recovery, mental health, and quality of life. Collectively, the economic burden of uncontrolled acute postoperative pain is vast, driven by significantly longer surgery center stays and higher rates of unplanned admissions and readmissions to emergency departments and hospitals [2].

An additional risk of poorly managed acute postoperative pain is the development of persistent postoperative pain, frequently defined as new and enduring pain of the operative or related area without other evident causes lasting more than 2 months after surgery. While prevalence of such “chronic” postsurgical pain (CPSP) varies by surgery type and generally decreases with time, it may occur in 10–60% of patients after common procedures [2,50,51,52,53]. The physical and mental consequences of persistent postoperative pain are frequently complicated by the development of persistent opioid use, which is also variably defined but largely refers to ongoing opioid use for postoperative pain in the timeframe of 90 days to 1 year after surgery [2,34]. The incidence of persistent postoperative opioid use appears highest after spine surgery and not uncommon (i.e., 5–30%) after arthroplasty and thoracic procedures. Patients on opioids prior to surgery demonstrate a 10-fold increase in the development of persistent postoperative opioid use. Still, previously opioid-naïve patients are converted to persistent opioid users by the surgical process at an alarming 6–10% rate [10,34]. Considering that 1 in 4 chronic opioid users may develop an opioid use disorder, the mitigation of persistent postoperative pain and opioid use should be a priority to healthcare providers and systems [10,54].

### 2.2. Opioid Stewardship, Multimodal Analgesia, and Equianalgesic Opioid Dosing

“Perioperative opioid stewardship” may be defined as the judicious use of opioids to treat surgical pain and optimize postoperative patient outcomes. The paradigm is not simply “opioid avoidance,” and requires balancing the risks of both over- and under-utilization of these high-risk agents. To this end, postoperative opioid minimization should be pursued only in the greater context of optimizing acute pain management, reducing adverse events, and preventing persistent postoperative pain through comprehensive multimodal analgesia [19,33,55,56,57,58,59,60,61]. Multimodal analgesia, or the use of multiple modalities of differing mechanisms of action, is key to decreasing surgical recovery times and complications, and so is also a fundamental component of the enhanced recovery paradigm promoted by the international Enhanced Recovery After Surgery (ERAS^®^) Society [19,24,62,63,64,65]. Dedicated resources and care coordination are often required for institutions to align analgesic use with best practices, so Opioid Stewardship Programs (OSPs) are taking hold, modeled after antimicrobial stewardship practices [29,38,66,67,68].

Quantifying opioid exposure for patient care, process improvement, or research purposes requires the use of a standardized assessment. Opioid doses can be normalized to their equianalgesic oral morphine amounts, i.e., Oral Morphine Equivalent (OME), oral Morphine Milligram Equivalent (MME), or oral Morphine Equivalent Dose (MED) [69,70,71]. Current evidence-based recommendations for equianalgesic dosing of opioids commonly encountered in perioperative settings are summarized in Table 1 [71]. Guidelines on the use of opioids for chronic pain are also available and provide slightly different conversions for MME doses, citing earlier literature [54,72]. All opioid conversions for patient care purposes should include careful consideration of the limitations of these factors, including extremely wide ranges for ratios found in clinical trials, clinical inter-patient variability, incomplete cross-tolerance between opioids, and other patient-specific factors (e.g., renal impairment or genetic variants in metabolism, see Section 3.5). The newly calculated opioid dose should therefore be reduced by 25–50% when changing between opioids or routes of administration, as discussed in detail elsewhere [71].

## 3. Pain Management and Opioid Stewardship across the Perioperative Continuum of Care

Perioperative care consists of a complex orchestra of medical professionals, physical locations, processes, and temporal phases. This continuum begins prior to the day of surgery (DOS), continues across inpatient or ambulatory stay, and extends through recovery and follow-up phases of care. A maximally effective institutional strategy for perioperative pain management and opioid stewardship includes all phases and providers across this continuum. Though there is no definitive evidence-based regimen, effective multimodal analgesia requires institutional culture and protocols for pre-admission optimization, consistent use of regional anesthesia, routine scheduled administration of nonopioid analgesics and nonpharmacologic therapies, and reservation of systemic opioids to an “as needed” basis at doses tailored to expected pain and preexisting tolerance [15,18,33]. Figure 1 summarizes the recommended strategies at each phase of care, which will be discussed in greater detail.

### 3.1. Pre-Admission Phase

The pre-admission phase of care occurs prior to the day of surgery (DOS) and represents the ideal opportunity for patient optimization. Safe and effective interventions exist during the pre-admission phase to improve pain control and decrease opioid requirements in the subsequent perioperative period. Recommended pre-admission interventions include evaluation of patient pain and pain history, education to patients and caregivers, assessment of patient risk for perioperative opioid-related adverse events (ORAEs) and implementation of mitigation strategies, optimization of preoperative opioid and multimodal therapies, and advance planning for perioperative management of chronic therapies for chronic pain and medication-assisted therapy for substance use disorders.

#### 3.1.1. Patient Pain History, Evaluation and Education

Perioperative pain management planning should be pursued through a shared decision-making approach and necessitates an accurate pre-admission history and evaluation. Pain assessment should include classification of pain type(s) (e.g., neuropathic, visceral, somatic, or spastic), duration, impact on physical function and quality of life, and current therapies. Other key patient evaluation components include past medical and psychiatric comorbidities, concomitant medications, medication allergies and intolerances, assessment of chronic pain and/or substance use histories, and previous experiences with surgery and analgesic therapies [15]. Barriers to the safe use of regional anesthetic and analgesic strategies can be identified and considered, such as certain anatomic abnormalities, prior medication reactions, a history of bleeding disorders, or need for anticoagulant use [73]. Likewise, chronic medications that synergize postoperative risks for ORAEs and complications can be managed expectantly, such as benzodiazepines (e.g., respiratory depression, delirium). While such medications may not be avoided feasibly due to the risk of withdrawal syndromes, consideration could be given to preoperative tapering and/or increased education and monitoring for adverse effects in the perioperative period [15,74].

Psychosocial comorbidities and behaviors that could negatively affect the patient’s perioperative pain management and general recovery include anxiety, depression, frailty, and maladaptive coping strategies such as pain catastrophizing [15,18,52,75,76,77,78]. Additionally, patients with chronic pain and/or history of a substance use disorder frequently experience anxiety regarding their perioperative pain management and/or risk of relapse [18]. While high-quality data is currently lacking to support specific pre-admission strategies for decreasing postoperative adverse events associated with mental health comorbidities, pilot studies and expert opinion support the integration of psychosocial optimization into the “prehabilitation” paradigm for surgical readiness [18,52,75,79]. Cognitive function, language barriers, health literacy, and other social determinants of health also significantly influence postoperative pain management and recovery [51,80,81,82]. Validated health literacy assessments have been applied to surgical populations [83,84,85,86,87]. Prospective identification of these challenges, including the application of standardized cognitive and psychosocial assessments, can allow for appropriate preoperative referral, patient optimization, and future study of risk mitigation strategies [15,18,52,75,78,80,88]. To this end, various predictive tools for postoperative pain are being explored [88,89,90,91].

Patient-centered education and expectation management during the pre-admission phase of care are effective strategies for improving postoperative pain control, limiting postoperative opioid use, decreasing complications and readmissions, and increasing postoperative function and quality of life [15,18,92,93,94,95,96,97,98]. Insufficient evidence exists to support specific educational strategies or components, but current guidelines recommend an individualized discussion about expected severity and duration of postoperative pain to generate realistic goals about pain management, a description of how pain will be assessed, and an overview of available analgesic options, including the judicious use of opioids and their associated risks, multimodal therapies in the form of nonopioid medications, local anesthetic or regional (central and peripheral) techniques, and nonpharmacologic modalities [15]. Patients with chronic pain or substance use disorders should especially be introduced to the concepts of multimodal analgesia and educated on the risks of perioperative opioids, beginning at the pre-admission phase of care [18]. Education should be provided in an effective manner considering the patient’s age, health literacy, language, and cognitive ability [15,99]. The patient’s prior experiences, preferences, and expectations should then be integrated into a collaborative, documented, goal-based plan [15].

Provider education, resources, and time constraints in pre-admission clinics currently limit the widespread uptake of these best practices into routine care. The pre-admission phase therefore represents an important target for process improvement related to perioperative pain management and opioid stewardship [94]. To support such efforts, some organizations have made patient education materials publicly available [100,101,102,103].

#### 3.1.2. Pre-Admission Opioid Use Assessment, Risk Stratification for Perioperative ORAEs, and Optimization

Recent guidelines have provided an updated tool recommended for preoperative opioid assessment, termed the Opioid-Naïve, -Exposed, or -Tolerant plus Modifiers (O-NET+) classification system (Table 2) [18]. Patients are deemed opioid-naïve if they have had no opioid exposure in the 90 days prior to surgery, opioid-exposed if they have taken any amount less than 60 milligram (mg) oral morphine equivalents per day (MED) in the same time period, or opioid-tolerant if they have taken 60 MED or more in the seven days before surgery. Risk modifiers are then utilized to stratify the patient’s risk for perioperative ORAEs, such as uncontrolled psychiatric disorders, any substance use disorder history, maladaptive behavioral tendencies that could impact pain management, and the surgical risk for persistent pain. These categories can then be used to guide perioperative risk mitigation strategies and optimization goals. Patients at every risk level benefit from preoperative education and expectation management in addition to multimodal analgesia throughout the perioperative care continuum. Additionally, patients at moderate risk for perioperative ORAEs should be referred for optimization of psychobehavioral comorbidities, and high risk patients should also be referred to a pain management specialist prior to surgery (Table 2). While not all identified risk factors may be modifiable in time for surgery, the O-NET+ classification system affords the ability to identify higher risk patients proactively to inform perioperative planning and support future practice research [18].

Patients using opioids prior to surgery should also receive a customized evaluation of their current analgesic regimen for optimization opportunities, which may include maximizing pre-admission multimodal therapies and/or tapering of opioid therapies. Conversely, certain pain medications may need to be interrupted for surgery (e.g., aspirin or other anti-inflammatory agents), in which case clinicians should provide clear rationale and education on safe resumption after surgery. Patients on long-term opioid therapies prior to surgery experience increased rates of postoperative complications in addition to higher rates of persistent postsurgical pain and prolonged opioid use, so preoperative opioid minimization has emerged as a potentially modifiable risk factor. To this end, current consensus statements and expert opinion suggest titrating preoperative opioid therapies to the lowest effective dose, depending on the patient’s underlying condition [18,104,105,106]. Patients currently taking more than 60 mg MED may be evaluated for a goal of tapering to less than this threshold by one week prior to surgery as a possible mechanism for reducing risk of perioperative ORAEs, since this should theoretically reduce postoperative opioid requirements. One study found similar postoperative outcomes between opioid-naïve patients and chronic opioid users who successfully reduced their preoperative opioid dose by at least 50% before surgery, and both of these cohorts experienced significantly improved outcomes compared to chronic opioid users who were unable to wean to this threshold [107]. Some experts have proposed delaying elective surgery in chronic pain patients for a structured 12-week prehabilitation program focused on opioid reduction (general goal of ~10% per week) and increasing psychological reserve ahead of painful procedures [108]. The ultimate goals of preoperative opioid minimization include improving postoperative pain control, limiting perioperative opioid exposure and associated ORAEs, and avoiding persistent dose escalations of chronic opioid therapies [18].

High-quality data does not exist at this time to support strong recommendations regarding preoperative opioid reduction strategies, so a patient-specific, collaborative approach informed by appropriate expertise is vital. General guidance exists for opioid tapering in patients on chronic opioid therapy, but application to the preoperative setting is not discussed [109,110]. Opioid tapering must always be accompanied by patient education and respectful support from the healthcare team [104,109]. Transitional pain services or other perioperative pain management specialist consultation is recommended for opioid-tolerant or otherwise high-risk patients by current guidelines and is supported by implementation reports [15,18,111,112,113,114]. Current institutional expertise and resources limit availability of such services at many centers, representing an important area for future investment by health-systems and institutions.

#### 3.1.3. Planning for Perioperative Management of Chronic Long-Acting Opioids and/or Medication Assisted Treatment (MAT)

Patients with chronic pain and/or substance use disorders pose significant challenges to perioperative pain management and opioid stewardship. These complex surgical populations are expected to continue growing, necessitating increased clinical knowledge and creativity from perioperative providers [115]. It is imperative that surgery centers create mechanisms for identifying these high-risk patients prior to surgery to allow for preoperative optimization and coordination of perioperative care. Pre-admission expert consultation is recommended, as is coordination with the patient’s chronic therapy prescriber, to allow for optimal perioperative care and safe transitions throughout the recovery period [15,18].

Perioperative management of chronic long-acting opioid receptor therapies, including those used as medication-assisted treatment (MAT) for substance use disorders, should be planned during the pre-admission phase of care. These high-risk medications include long-acting pure mu-opioid receptor agonists (e.g., OxyContin^®^), methadone, a multitude of buprenorphine products, and the pure opioid antagonist naltrexone (Table 3). A thorough pre-admission medication reconciliation is imperative, including the assessment of available prescription drug monitoring program (PDMP) data, since the use of these products span many formulations and therapeutic indications that may not be evident upon history and physical alone. For example, buccal, transdermal, and implanted formulations of buprenorphine are increasingly used for chronic pain indications. Additionally, naltrexone is used off-label for self-mutilation behavior, and is also available in a combination oral product labeled for weight management (Contrave^®^). Table 3 summarizes current general recommendations for perioperative management of chronic opioid receptor therapies.

Chronic pain and opioid tolerance are frequently complicated by opioid-induced hyperalgesia, physical dependence, psychological comorbidities, and/or substance use disorders, making postoperative pain more difficult to manage in this population [104,116,117,118]. These factors contribute to current expert recommendations to continue chronic long-acting opioid agonists throughout the perioperative period, including methadone and buprenorphine [18,115,116,119,120,121,122]. Methadone and buprenorphine can be prescribed for either chronic pain treatment or as medication-assisted treatment for opioid use disorder (OUD) in the outpatient setting.

Conventional belief has been to discontinue buprenorphine therapy prior to surgery to allow for unencumbered mu-opioid receptors and more effective perioperative analgesia. Current data and clinical experience have challenged this practice, and experts cite multiple reasons for supporting perioperative continuation over interruption. Firstly, buprenorphine is now better understood as an efficacious analgesic, and likely one without ceiling dose effect for analgesia. Little data exists to support better pain control with buprenorphine cessation. Ceiling effects are observed for respiratory depression and sedation, however, likely conferring a safer risk profile than pure mu-opioid agonists [104,122,129,130,131,132]. Buprenorphine has also demonstrated protective effects against opioid-induced hyperalgesia, likely improving postoperative pain responsiveness to therapy [121]. This notion is supported by retrospective evidence that chronic buprenorphine users exhibit lower postoperative opioid requirements when buprenorphine is given on day of surgery versus when it is not [133]. These unique qualities suggest buprenorphine continuation is beneficial to pain control and opioid safety in the perioperative period, and preoperative cessation of therapy removes these benefits when they may be most advantageous. A more nuanced strategy is to temporarily increase and/or divide buprenorphine or methadone dosing starting on the day of surgery to maximize pain control without increasing peak-related adverse effects. This has pharmacologic merit in that the analgesic duration of action for buprenorphine and methadone is far shorter than their active duration for reducing cravings [121,128].

For patients on buprenorphine doses exceeding 8–12 mg/day, some experts consider a preoperative reduction to 8–12 mg/day that is then continued throughout the perioperative period, in concert with the patient and buprenorphine prescriber [122,126,132] (see also Section 3.5.3). Data describing the impact of this strategy on patient-centered outcomes remains limited. An alternative option that has previously been proposed is transitioning the patient to a pure mu-opioid agonist (e.g., methadone) prior to surgery. This strategy creates challenges when converting back to buprenorphine postoperatively due to the risk of precipitous withdrawal and length of time (days) involved. Additionally, removing the protective effects of partial agonism to overdose risk likely makes this strategy less safe, and we discourage its use [123].

Preoperative discontinuation of buprenorphine is no longer recommended [18,119,120,122,126,132]. Complete buprenorphine cessation can lead to opioid withdrawal syndrome if sufficient alternative opioid agonists are not administered, and standard perioperative protocols may not be adequate for this purpose. While not life-threatening, opioid withdrawal is physically and psychologically taxing to the patient and is likely to contribute to increased perioperative opioid exposure, postoperative complications, prolonged hospital stays, and increased healthcare costs. In addition to necessitating increased doses of less safe opioids for adequate postoperative pain control, interruption of chronic buprenorphine therapy requires a subsequent opioid-free period prior to reinitiation. This is especially problematic in a population that may be experiencing opioid-induced hyperalgesia, uncontrolled pain, unmet psychosocial needs, continuity of care gaps, and access to non-prescribed opioids in the postoperative period. While clinical data is limited, expert opinion cites this dynamic as a key driver of postoperative opioid misuse and opioid use disorder development or relapse [74,119,120,122,123,126].

In short, buprenorphine is appropriately viewed as an effective basal analgesic therapy with possible protective effects against ORAEs, psychological destabilization, and relapse. Therapy interruption at the time of painful stimulus is likely to exacerbate the underlying indication for buprenorphine, opening the door to inadequate pain control, increased postoperative complications and costs, and opioid misuse. To this effect, a recent clinical practice advisory states, “it is almost always appropriate to continue buprenorphine at the preoperative dose; furthermore, it is rarely appropriate to reduce the buprenorphine dose” [119]. This is supported by current consensus statements and expert reviews [18,120,121,122,123,124,125,126,127,128]. Rigorous evidence on postoperative pain management in patients on MAT remains urgently needed to quantify these anecdotal benefits and to compare the effects of available perioperative strategies on patient-centered outcomes [115]. It is also important for healthcare providers to understand the role of buprenorphine coformulation with naloxone, and that continuing combination products (i.e., Suboxone^®^) poses no risk of opioid reversal when the dosage form is taken appropriately. The naloxone is only made bioavailable when the dosage form is altered in an attempt to inject it, and was developed as an abuse deterrent [126].

Conversely, naltrexone formulations must be discontinued in sufficient time to ensure complete wash-out prior to surgery to avoid iatrogenic pain crisis, since opioids are rendered largely ineffective during therapy [123,124]. Animal data suggest opioid therapies would need to be increased 10–20 times the standard clinical dose to achieve analgesia in patients on concomitant naltrexone [134], and human data is very limited [115,135]. Chronic naltrexone therapy induces opioid receptor up-regulation, however, so patients usually on naltrexone therapy may exhibit increased sensitivity to opioids after naltrexone discontinuation for surgery [117,136]. Postoperative planning for such patients should include maximal nonopioid therapies, opioid-naïve dosing for as-needed opioids, and increased monitoring for adverse events [117,124,128,135].

#### 3.1.4. Perioperative Planning for the Patient with Active Substance Use

A thorough social history is imperative to proactively identifying other substance use that may have significant consequences for postoperative pain management. Patients who exhibit misuse of prescription and/or illicit opioids and also require surgery pose an exceptional challenge [137]. Providers should anticipate postoperative withdrawal symptoms and increased pain sensation in patients with active opioid use disorder (OUD) and ensure postoperative monitoring using validated measures [123,128,138]. Perioperative planning should include opioid withdrawal management and maximizing multimodal agents, including ketamine [104,123,139,140]. Medication-assisted treatment (MAT) initiation and optimization of psychiatric comorbidities should be attempted in the pre-admission phase when time and patient desire allow. If MAT initiation is not possible or desirable prior to surgery, planning for postoperative inpatient MAT initiation should be pursued, with patient consent. This should involve consultation with the inpatient addiction medicine consultant, who will also arrange outpatient follow-up and post-discharge resources for continued OUD management [123].

Patients with alcohol use disorder should be managed expectantly in the postoperative period using validated assessments [141,142]. While such patients do not demonstrate cross-tolerance requiring increased opioid doses to effectively treat pain, the concomitant use of benzodiazepines will confer an increased risk of respiratory depression and increased monitoring is needed. Likewise, patients using prescribed or illicit benzodiazepines should not be prescribed higher than routine opioids for postoperative pain, but are subject to increased postoperative respiratory risk [140,143]. Increased opioid tolerance has also not been observed in postoperative patients with baseline cocaine and/or amphetamine use, but stimulant withdrawal can occur upon cessation that may add to postoperative anxiety and discomfort [140].

Recreational and medicinal cannabinoid use is expanding, including various applications to chronic pain management, and may be replacing chronic opioid and other substance use in some patients [144,145,146]. Providers should actively engage patients in shared decision-making and education regarding the perioperative implications of chronic cannabinoid use (discussed comprehensively elsewhere [147,148]), including how postoperative pain is affected. Cannabinoid use is associated with significantly increased anesthetic requirements during surgery, higher postoperative pain scores, higher perioperative opioid consumption, and poorer postoperative sleep quality [149,150,151,152]. This may be due to cannabinoid receptor downregulation and the complex interactions of the endocannabinoid system with various neurotransmitters and pain modulation pathways [153,154]. Cannabinoids may also increase risks for perioperative medical complications and drug interactions, and so many practitioners are advising perioperative cessation [148]. Chronic cannabinoid users will experience an uncomfortable withdrawal syndrome after abrupt cessation, however, so preoperative down-titration and close postoperative monitoring may be considered [104,140,155]. High-quality evidence to guide perioperative management of active substance use remains elusive.

### 3.2. Preoperative Phase

The preoperative phase of surgical care begins at patient presentation to the preoperative area on the day of procedure (“postoperative day zero” or POD0). This onsite period, prior to the administration of sedatives or anxiolytics, is ideal to renew education and expectation-setting regarding perioperative analgesia. The patient and caregiver(s) should be engaged in shared decision-making to finalize the anesthetic plan and complete consent documentation.

Preoperative anxiety is common among patients and caregivers. Patient education is associated with decreased anxiety, and nonpharmacologic modalities improve relaxation and positive thinking as part of a multimodal approach to postoperative pain management [15]. While evidence is insufficient to strongly recommend specific strategies, perioperative cognitive-behavioral therapies including guided imagery and music therapy are noninvasive and unlikely to cause harm. Their positive effects on reducing anxiety may provide downstream benefits to narcotic avoidance and analgesia, but further study is needed [15,55,156,157,158,159,160]. Massage and physiotherapy have contributed to improved pain control in other settings and are being explored for perioperative applications [55]. Preoperative virtual reality technology has also been successfully employed to reduce perioperative anxiety and pain [161,162,163].

Most notably, the preoperative phase of care should be employed to administer preemptive analgesia. Preemptive analgesia refers to the administration of analgesics *prior* to a painful stimulus (i.e., surgical incision) to decrease subsequent pain response. A complex interplay between surgical incision and preexisting factors drives a cascade of central and peripheral sensitization, inflammation, and neuromodulation that intensifies and prolongs postoperative pain beyond the point of physical healing. Preemptive analgesia attenuates these processes to confer reduced postoperative pain, decreased opioid requirements, and potentially less-frequent development of persistent postsurgical pain across diverse procedures [15,53,164,165,166,167,168,169,170,171,172]. Preemptive analgesics can generally be administered orally with sips of water one to two hours prior to operating time. This strategy is expected to maximize efficacy by aligning pharmacokinetics with therapeutic goals and avoids the risks and costs of unnecessary intravenous agents, which are unlikely to confer meaningful benefit over their enteral counterparts [15,169,173,174,175,176]. Intravenous agents should be employed in patients with true contraindications to enteral administration or in those with significantly impaired enteral drug absorption. 

While every surgical patient should be offered multimodal preemptive analgesia as a component of comprehensive perioperative analgesia and opioid stewardship, not every patient is an ideal candidate for each medication. Table 4 contains a sample preemptive analgesia protocol with applicable patient-specific exclusion criteria. The optimal pharmacologic agents and doses for preemptive analgesia are undetermined. Acetaminophen is frequently used alongside anti-inflammatory and neuropathic agents, and the combination of these three classes appears to provide the greatest opioid-sparing benefit [177]. Preemptive acetaminophen should be employed widely due to its favorable safety profile, including in patients with cirrhosis [178]. Preemptive opioids may be counterproductive, however, even in opioid-tolerant patients, and are not recommended preoperatively [15,18,106,179]. Preemptive opioids should be especially avoided in opioid-naïve patients due to the risk of increasing postoperative pain perception and opioid use [180].

The use of perioperative gabapentinoids has been increasingly controversial owing to conflicting evidence of analgesic benefit and risks of adverse effects, including dizziness and synergistic sedation with concomitant opioids [61,185,186,187,188,189,190]. The U.S. FDA has issued additional warnings regarding the risk of respiratory depression with gabapentinoids in patients who have respiratory risk factors, including the elderly, the renally impaired, those with chronic lung diseases, and those on concomitant sedatives [191]. This warning cited predominantly observational data and emphasized the need for patient-specific risk assessments. One of the reviewed studies suggested increased risk with preoperative gabapentin doses over 300 mg [61], while another did not identify any significantly increased risk when exposure was limited to a single preoperative dose [189]. A third retrospective analysis found preoperative gabapentin exposure was associated with a 47% increase in odds of experiencing a postoperative respiratory event, though the vast majority of the studied population were administered doses exceeding 300 mg [190,191]. Gabapentinoids exhibit dose-dependent propensity to increase postoperative pulmonary complications, though combination with other multimodal agents may negate this risk, and the absolute risk of adverse events with perioperative gabapentinoids appears low [177,192,193]. Hence, adverse event risks of gabapentinoids can be substantially mitigated by using conservative doses (i.e., 300 mg gabapentin preoperatively), avoiding postoperative use in patients experiencing or at risk for sedation or dizziness, and/or avoiding entirely in high-risk patients. 

Despite these limitations, gabapentinoids have consistently demonstrated significant opioid-sparing benefits and reduced postoperative nausea [15,60,185,194,195,196,197,198,199]. A recent meta-analysis suggested minimal analgesic benefit to perioperative gabapentinoids in terms of patient-reported pain scores, yet found a significant opioid reduction of approximately 90 mg oral morphine over the first seventy-two postoperative hours [185]. Additionally, gabapentinoids may mitigate central sensitization and decrease the risk of persistent surgical pain, though further research is needed [53,172,200]. Opioid-tolerant patients may especially benefit [117]. Hence, gabapentinoids remain a valuable tool in the perioperative opioid stewardship arsenal for appropriate patients and are supported by multiple guidelines [15,18,197,201]. Ongoing controlled trials may further delineate the effectiveness, safety, and cost-effectiveness of perioperative gabapentinoids [202].

Some pharmacokinetic differences exist between gabapentin and pregabalin, though both are heavily renally eliminated. Pharmacokinetic profiling suggests an equipotent ratio of 6:1 for gabapentin:pregabalin doses [203]. Some have suggested that switching to pregabalin from gabapentin may reduce adverse events in the chronic neuropathic pain setting, but these benefits were not sustained or significantly different from patients who remained on gabapentin [204]. The relative safety profiles of the gabapentinoids in perioperative settings are therefore unlikely to differ when use is limited to short-term, low doses. Duloxetine, a serotonin- and norepinephrine-reuptake inhibitor with analgesic properties, has also been effective in perioperative multimodal regimens, representing a potential alternative to gabapentinoids [205,206,207,208,209,210].

Nonsteroidal anti-inflammatory drugs (NSAIDs) have long been shrouded in safety concerns of variable validity [183]. Bleeding risk has been of primary concern with perioperative NSAID exposure given the anti-platelet effects of cyclooxygenase-1 (COX-1) inhibition. Bleeding times and postoperative bleeding events do not appear significantly affected by NSAIDs at usual doses, and this risk may be further mitigated by using COX-2 selective agents [211,212,213,214,215,216]. Traditional dogma has suggested avoiding NSAIDs in spinal/orthopedic fusion surgeries because of the risk of nonunion. More recent and higher quality data suggests short-term NSAID use at normal doses does not affect spinal fusion rates and is valuable for postoperative analgesia and opioid minimization [60,167,217]. High-quality prospective studies are needed to definitively assess this risk. In gastrointestinal surgery, NSAID use has been associated with increased risk of anastomotic leak, but recent meta-analyses suggest this concern may be limited to non-selective NSAIDs [218,219,220].

Available literature suggests celecoxib, a selective COX-2 inhibitor, is not associated with the aforementioned concerns with NSAID use in spine and gastrointestinal surgery [60,218,219,220]. Celecoxib is the only NSAID specifically recommended for preoperative use in clinical practice guidelines for postoperative pain management, likely owing to the significant evidence in this setting and lower rates of some adverse effects [15,212]. While celecoxib could be viewed as the NSAID of choice for perioperative use in many surgical populations, it must be avoided in cardiac surgery, where selective COX-2 inhibitors have been associated with increased rates of major adverse cardiac events [201,221]. Increased rates of adverse cardiac events have not been demonstrated with nonselective NSAIDs in cardiac surgery, nor with selective COX-2 inhibitors in noncardiac surgery [183,222]. Caution may still be warranted with selective COX-2 inhibitors in noncardiac surgery patients with significant cardiovascular disease, but these risks may not be significant when exposure is limited to short-term perioperative use [183,212,223,224,225]. Patient-specific risk-benefit assessments regarding perioperative NSAID use are warranted and should include consideration of the risks of increased pain and opioid use in each given patient [183]. All perioperative NSAIDs are inadvisable in patients with preexisting renal disease or otherwise at high risk of postoperative acute kidney injury [226,227,228,229,230]. NSAIDs, including celecoxib, should not be withheld in patients with sulfa allergies, however [231,232,233]. Although chronic NSAID should be avoided in bariatric surgery patients, short-term perioperative use is considered safe and beneficial, and is recommended in this population per current guidelines [234,235,236]. Concomitant, temporary proton pump inhibitor therapy could be considered in patients with high gastrointestinal risk.

### 3.3. Intraoperative Phase

Anesthetists are crucial team members in optimizing perioperative pain management and opioid stewardship since these aspects, alongside many postoperative outcomes, hinge upon effective anesthesia. Anesthetic strategies include general, regional, and local modalities, as reviewed comprehensively elsewhere [237,238,239,240,241]. General anesthesia has progressed from its origins in deep, long-acting sedative-hypnotics to a more “balanced” strategy employing a combination of agents to create the anesthetized state while facilitating quicker recovery. Balanced general anesthesia now includes broader multimodal agents to mitigate surgical stress and decrease reliance on systemic opioids [242]. Regional anesthesia is divided into neuraxial and peripheral strategies, and various techniques within these strata are reviewed (Table 5). These ever-expanding anesthetic options have rendered controlled comparative efficacy studies challenging, limiting available guidance on optimal techniques for perioperative analgesia and opioid stewardship. Furthermore, the feasibility of anesthetic strategies varies widely by procedure type, anesthetist training, institutional capabilities, and patient-specific factors. Multiple professional collaboratives have generated quality procedure-specific reviews and recommendations to which perioperative teams should refer when developing anesthetic pathways at the institutional level [20,22].

#### 3.3.1. Regional and Local Anesthesia

Regional anesthesia is a cornerstone of multimodal analgesia and opioid minimization, in addition to reducing perioperative morbidity and mortality. General anesthetics can be reduced or sometimes avoided with regional anesthesia, resulting in shorter recovery times and less adverse drug effects such as postoperative nausea and vomiting. Hence, regional anesthesia is integral to the enhanced recovery paradigm [23,62,63,243,244,245]. The benefits of regional anesthesia continue to be explored and include reduced cancer recurrence when used in oncologic surgeries, likely owing to the mitigation of inflammatory marker surges and other immunomodulatory effects [246,247]. While regional anesthesia is a foundational modality for perioperative analgesia and opioid stewardship, it requires input from patients, expertise from clinicians, and careful procedural assessment and institution-specific tailoring of anesthetic options [15,62,63,248]. Key components and considerations for regional and local anesthetic strategies are summarized in Table 5.

The main limitation of local anesthetics is their duration of action, which diminishes their ability to provide opioid-sparing analgesia for multiple postoperative days [249]. One strategy for extending clinical duration of regional anesthesia is the addition of pharmacologic adjuvants such as dexamethasone, clonidine or dexmedetomidine, and/or epinephrine [249,250,251,252,253,254]. While additives to local anesthetics may extend duration of peripheral nerve blockade by as much as 6–10 h and are supported by clinical practice guidelines, total duration of action for single-shot injections will still be limited to less than 24 h [15,249,252]. Additionally, despite considerable research, data remains of low quality and with conflicting results for common pharmacologic adjuvants to peripheral nerve blocks, and they may confer additional risks. These dynamics preclude strong recommendations or expert consensus regarding their use [251,252]. Alternatively, continuous catheters are effective strategies for extending local anesthetic analgesia, and are supported by clinical practice guidelines when the duration of analgesia is expected to exceed the capacity of single-injection nerve blocks [15,255,256]. Continuous catheters are not without limitations, however, including increased complexity to perform and maintain, catheter-related complications, and additional monitoring and follow-up requirements [249]. As such, controlled-release local anesthetic formulations have also been developed [257,258,259]. Liposomal bupivacaine has not demonstrated clinically meaningful benefits to postoperative pain control or opioid reduction when compared to conventional local anesthetics in local wound infiltration, periarticular injection, or peripheral nerve blockade [249,260,261,262,263,264,265,266,267,268,269,270,271,272,273,274,275]. Potential benefits and cost-effectiveness of extended-release local anesthetic formulations are likely to vary significantly depending on injection technique, site, and type of surgical procedure, so institutions should consider surgery- and patient-specific use of these agents.

To ensure patient safety, it is imperative to have a standardized, collaborative assessment of the total local anesthetic exposure from all sources. Clinicians must remain vigilant to ensure toxic doses are not reached inadvertently when using multiple local anesthetics across anesthesia and surgical applications (i.e., peripheral nerve block in addition to periarticular injection in total knee arthroplasty). Furthermore, local anesthetic toxicity may be masked while a patient is under general anesthesia. To avoid cardiovascular collapse and death, local anesthetic systemic toxicity must be recognized and treated early [276,277]. Accordingly, current guidelines recommend against intravenous lidocaine within four hours of most local anesthetic-containing regional anesthetic strategies, though local anesthetic infusions through wound or epidural catheters may be started without boluses at thirty minutes after IV lidocaine has been stopped [26]. Additionally, local anesthetics must be used extremely carefully in patients with Brugada Syndrome due to potential arrhythmic effect [278].

**Table 5 healthcare-09-00333-t005:** Selected Attributes of Regional and Local Anesthetic Strategies for Pain Management and/or Opioid Stewardship.

Category, General Considerations	Anesthetic Strategy	Application	Specific Clinical Considerations
**Neuraxial Regional Anesthesia** Provides motor, sensory, and sympathetic blockade Includes local anesthetics +/− opioids May serve as primary or adjunctive anesthetic or analgesic strategy Significantly improves pain control and decreases use of systemic narcotics May decrease postop morbidity and mortality Increases risks of urinary retention, hypotension Rare catastrophic complications Requires interruption and careful management of antithrombotics			
	Spinal (intrathecal) injections	Single injection of local anesthetic +/− opioid ^1^ into subarachnoid space; for surgeries below umbilicus	Hypotension, pruritus (if opioid used); Requires careful assessment and monitoring of postop narcotics if opioid used
	Epidural infusions	Continuous infusion +/− PCEA or PIEB of local anesthetic +/− opioid into posterior epidural space; wide range of procedures (thoracic, abdominal, lower extremity)	Infusion pumps and catheters require special monitoring; may complicate or delay postop mobility or pose other logistical challenges; require careful postop narcotic management if opioid used
	Para-vertebral blocks	Single/multiple injections or catheter placement for continuous local anesthetic infusion along vertebra near spinal nerve emergence; for thoracic or abdominal procedures	Effective blockade of complete hemithorax or hemiabdomen but technically difficult; modern practice generally favors fascial plane blocks or alternative neuraxial modalities
**Peripheral Regional Anesthesia** Includes local anesthetic injections or infusions (CRA), +/− pharmacologic adjuvants Can limit/avoid need for general anesthesia for some procedures, or can be combined with anesthesia as analgesic strategy Fewer risks and contraindications than neuraxial techniques as most are IM injections Most do not provide sympathetic block Significantly improves analgesia, decreases narcotic requirements May decrease morbidity Rare risks of nerve injury, bleeding, infection, LAST Use of ultrasound guidance has increased safety and consistency			
	Plexus blocks	Brachial plexus blocks for unilateral upper extremity procedures; lumbar plexus blocks for hip or lower extremity	Requires significant clinician expertise of anatomy; proximal brachial plexus blockade risks hemidiaphragmatic paresis
	Peripheral nerve blocks	Provide targeted anesthesia and/or analgesia of specific nerve or nerve bundles for extremity procedures	Numb limb or distribution must be protected from inadvertent injury, such as thermal injuries, hyperextension, or falls
	Fascial plane blocks (e.g., TAP, ESPB, FIB, PECS-2)	Use higher volumes of dilute local anesthetics to target dermatomes/nerve planes; for thoracic, abdominal, spinal or extremity procedures	Provide unilateral, dermatomal, or regional analgesia; increasing use in modern practice due to safety, ease of administration and broad applications
	Intravenous blocks (IVRA)	Use high doses of short-acting local anesthetic injected into venous system of an exsanguinated distal extremity to provide anesthesia and analgesia	High doses of local anesthetic are used so dual tourniquets must be used and their release carefully timed to prevent LAST; use limited to procedures less than 1 h
**Local Anesthesia** Mild sensory blockade of superficial/cutaneous nerves Minimal side effects Caution with type of local anesthetic, total exposure, and comorbid conditions (e.g., Reynaud) Avoid open wounds and compromised dermis with some techniques/products			
	Wound infiltration	SC and/or intradermal injection(s) by surgeon for incisional pain	Less effective if injected into areas of tissue infection
	Periarticular injections	Generally injected by surgeon without use of ultrasound guidance, such as in TKA	Provides effective postop analgesia, in some cases minimizing the need for peripheral nerve blockade
	Topical	Applied as sprays, creams, gels, patches, or oral rinses for superficial pain	Some can be safely self-administered by patient

^1^ Routine intrathecal opioids are not recommended by some guidelines [188]. Abbreviations: CRA = continuous regional anesthesia, ESPB = erector spinae plane block, FIB = fascia iliaca block, IM = intramuscular, IV = intravenous, IVRA = intravenous regional anesthesia (e.g., Bier block), LAST = local anesthetic systemic toxicity, PECS-2 = pectoralis nerve block (2 injections), PCEA = patient-controlled epidural analgesia, PIEB = programmed intermittent epidural bolus, SC = subcutaneous, TAP = transversus abdominis plane block, TKA = total knee arthroplasty. References: [15,18,23,170,188,237,240,242,249,250,255,279,280,281,282,283,284,285,286,287].

#### 3.3.2. Systemic Multimodal Adjuncts

Limitations to regional anesthesia include patient and systems factors. As such, systemic multimodal adjuncts should be implemented or used concurrently with regional anesthesia. These systemic therapies are usually started perioperatively and limited to the intraoperative phase of care or continued into the short-term recovery or postoperative phases. Table 6 summarizes dosing and clinical considerations for common intraoperative multimodal analgesics administered systemically.

Lidocaine infusions are one adjunct that may be applied in the perioperative period. Data exist for lidocaine infusions as opioid-sparing modalities across multiple procedure types, though most literature is for intra-abdominal procedures. Multiple studies have suggested decreased pain scores, decreased 24-h postoperative opioid usage, possible decreased length of stay, and minimal adverse effects [15,18,26,281,288,289,290,291]. Studies vary widely regarding the dosing of lidocaine infusions, whether or not boluses are administered, and infusion duration [291,292,293,294]. Although lidocaine infusions are frequently started intraoperatively, some centers may instate or continue therapy in the postoperative period where supported by institutional protocols [290]. Lidocaine infusions have been used to provide analgesia outside of the surgical arena, such as in patients with traumatic rib fractures [295]. Current guidelines generally recommend a loading dose of no more than 1.5 mg/kg be given as an infusion over 10 min, followed by an infusion of no more than 1.5 mg/kg/h for no longer than 24 h [26]. All doses must be calculated based upon ideal body weight and should not exceed 120 mg/h in any patient. Doses should be substantially reduced in patients with mild renal or hepatic dysfunction, and avoided entirely in patients with moderate or significant end organ dysfunction and in those weighing less than 40 kg. Other relative contraindications should be evaluated prior to use, including cardiac disease, electrolyte disorders, seizure and other neurologic disorders, and pregnancy or breastfeeding. Serum lidocaine level monitoring is not generally warranted with short-term perioperative use but could be considered if toxicity concerns emerge. Extensive monitoring recommendations should be reviewed and standardized institutional protocols put in place for this modality [26,296].

Similarly, sub-anesthetic ketamine by bolus or infusion has been applied to perioperative and inpatient settings for nonopioid analgesia. Ketamine’s ability to improve analgesia and mitigate opioid tolerance and hyperalgesia stems from its antagonism at the NMDA receptor; however, ketamine has a complex receptor profile that likely informs multiple acute and chronic pain pathways. While ketamine may be appropriately considered for opioid-naïve patients undergoing painful procedures, it is especially beneficial to the opioid-tolerant population [15,18,25,117]. Professional consensus statements exist for both intravenous lidocaine and ketamine use for postoperative analgesia and should be consulted. Patient selection, monitoring, and systems implementation are imperative for safety and success with these agents [25,26]. 

Magnesium has been investigated for its role in attenuating acute and chronic pain. Proposed mechanisms include magnesium’s antagonism of the NMDA-receptor, similar to that of ketamine. NMDA-receptor antagonism may interrupt central sensitization of pain, therefore allaying the pathologic transition from acute to chronic pain. An additional potential mechanism is magnesium’s antagonistic effects on calcium, as elevated levels of calcium are involved in central sensitization [297,298,299,300].

Other systemic medications studied for nonopioid perioperative analgesia include the α_2_-adrenergic receptor agonists dexmedetomidine and clonidine. These medications provide central analgesia and decrease agitation and sympathetic tone without significant inhibition of respiratory drive. Dexmedetomidine is a highly selective agonist at the α_2_-2A receptor subtype, which mediates analgesia and sedation from multiple locations within the central nervous system. This central sympatholysis blunts surgical stress and decreases kidney injury, though evidence is limited [261,317,320,321]. Similarly, esmolol has been investigated as a synergistic analgesic intraoperatively. Esmolol may contribute to antinociception by blunting sympathetic arousal transmission through β-adrenergic receptor antagonism, but mechanisms and benefits are still being elucidated [324,325].

Systemic multimodal analgesics have been studied as additives to peripheral and/or neuraxial regional anesthetic strategies, including magnesium, α_2_-agonists, dexamethasone, and methadone. Limited comparative efficacy among routes of administration has emerged. This appears most true for dexamethasone, which confers similar benefits to pain control and opioid use when administered via either modality [259,327,328,329,330,333]. Although administering dexamethasone as a component of peripheral nerve blockade may avoid systemic side effects, perineural dexamethasone may have a local effect on nerve tissues that may be undesirable in some patient populations. While literature exists for individual additives to various regional anesthetic techniques, there is no widely accepted consensus regarding ideal drug selection and dosing and if/when systemic administration is preferred [15,250,254,259,300,331,332,341].

Methadone is a systemic multimodal agent explored with increasing interest. A unique opioid in kinetic and mechanistic properties, methadone can be administered once intravenously at procedure commencement to provide prolonged analgesia into the postoperative period. In addition to mu-opioid receptor agonism, methadone’s complex mechanism includes NMDA-receptor antagonism and inhibition of serotonin and norepinephrine uptake in the central nervous system. These actions confer benefit in the treatment of chronic neuropathic pain and may also inhibit surgical stress and central sensitization, thus reducing the risks of opioid-related hyperalgesia, tolerance, and persistent postoperative pain [335,336,337,339,342,343]. Appropriate monitoring and communication across transitions of care is important when the anesthetist administers methadone intraoperatively. Education and processes should be implemented to ensure reduced subsequent opioid use and minimization of ORAEs, especially the risk of respiratory depression with concomitant narcotics given during methadone’s prolonged and variable half-life. Alerts embedded in the medication administration record may be ideal, since a “once” dose of intraoperative methadone is likely to be missed by providers in subsequent phases of care, despite its ongoing medication effects in the patient. Still, methadone appears a viable option in the multimodal arsenal and likely a preferable alternative to some clinicians’ use of long-acting pure opioids (e.g., OxyContin^®^) in preemptive protocols.

Systemic multimodal agents available to the intraoperative phase of care are plentiful but remain underutilized. This phenomenon results from the lack of high-quality data to guide many patient care decisions, especially comparative efficacy to inform agent selection, dosing, combination, and contraindications. Institutions are encouraged to generate collaborative protocols and processes that support the safe use of these agents in appropriate patients, including pre-built order sets with recommended patient selection, drug dosing, and monitoring. Deciding and designing an institution-specific “menu” of supported intraoperative options with appropriate safeguards should increase practice utilization and research opportunities.

### 3.4. Recovery Phase

Ample research supports preoperative nerve blocks to facilitate quicker discharge from post-anesthesia care units (PACUs), owing to their opioid-sparing properties and associated reductions in ORAEs, especially postoperative nausea and vomiting. Patients who undergo surgical procedures with nerve blocks as their primary anesthetic may bypass PACU Phase I with a quicker discharge, enabling increased throughput and efficiency of care while maintaining patient safety and opioid stewardship [63,255,261,344,345].

Multimodal and opioid-sparing strategies should be continued while a patient is in the recovery phase. However, when continuing multimodal strategies, clinicians must be mindful of prior doses of similar agents administered in prior phases of care. When patients are sufficiently awake, providers should limit the intravenous route of opioid administration per current guidelines [15]. Oral administration facilitates longer analgesia with fewer peak-related adverse effects and risks as compared to intravenous routes. Sublingual administration of concentrated oral opioid preparations may be an advantageous strategy for increasing onset of analgesic action with fewer risks than the intravenous route, but this warrants additional study [346]. Additionally, nonpharmacologic analgesic and anxiolytic strategies should be reintroduced in the recovery phase to facilitate patient comfort without reliance on narcotics [158,159,160,347,348,349,350,351,352].

Deliberate opioid stewardship, avoidance of the IV route of administration, and maximal multimodal analgesics are also crucial for facilitating timely discharge from PACU for same-day surgical patients. Regional anesthesia and lighter levels of intraoperative sedation, combined with more minimally invasive surgical techniques, are allowing many previously inpatient procedures to be pursued in the ambulatory setting [353,354,355].

### 3.5. Postoperative Phase

Postoperative pain management should be individualized to the needs of each patient, noting goals and response to the prescribed approach. This requires the use of a validated pain assessment tool (e.g., numerical, verbal, or faces rating scales, or visual analog score) to assess pain intensity on a recurring basis in addition to functional assessments and evaluation for adverse events [15]. Additionally, pain assessment tools should be appropriate for the patient’s age, language, and cognitive ability [15]. The pain assessment should be made during movement as well as at rest, and must include location, onset and pattern, quality or type of pain (i.e., nociceptive, visceral, neuropathic, or inflammatory), aggravating factors, and response to treatment. Typically, assessments should be performed 15–30 min and 1–2 h after administration of parenteral and oral analgesics, respectively, and less frequently for patients with stable pain control. However, analgesic regimens should not be adjusted based on pain ratings alone, given their inherent limitations for predicting analgesic requirements and the increased risk for opioid overexposure [356,357,358,359]. Functional assessment of how pain is influencing the patient’s ability to achieve postoperative recovery goals should be integrated into a multidimensional approach to adjusting therapeutic regimens [360,361]. Providers should also use pain assessment interactions to reinforce realistic expectations and include the patient in treatment plans throughout the hospital stay. Providers should also be mindful of implicit bias risks when assessing and treating pain. Multiple analyses have found that lower amounts of analgesics are routinely prescribed to Black and other patients of color despite higher degrees of self-reported pain, and that race influences prescriber perceptions of risk for opioid misuse [362,363,364].

Many of the strategies discussed herein for inpatient postoperative patients may also be applied to various special populations, including trauma/emergent surgical patients, the elderly, the obese, obstetric populations, and pediatrics, as discussed in more detail elsewhere [293,300,365,366,367,368,369,370,371,372,373,374,375,376,377].

#### 3.5.1. Postoperative Nonopioid Considerations

Postoperative pain management should continue to incorporate multiple treatment modalities to maximize therapeutic benefits and minimize complications, including nonpharmacologic strategies (Table 7) [15,55]. Physical modalities, including transcutaneous electrical nerve stimulation (TENS), acupuncture, massage, or cold therapy, alone or in combination with medications, may offer pain relief and reduce opioid use, though evidence is variable [15,55,158,160,347,350,378]. Preliminary evidence also suggests cognitive behavioral therapy (CBT), acceptance and commitment therapy (ACT), other mindfulness-based psychotherapy and music may reduce postoperative pain intensity and disability [15,79,379,380,381]. Surgery centers should devote due resources to making a variety of nonpharmacologic therapies standardly available to postoperative patients, as strongly supported by current guidelines and regulatory requirements [15,18,36].

To provide effective multimodal and opioid-sparing analgesia, clinicians should standardly provide around-the-clock nonopioid medications after surgery [15,18,33]. Acetaminophen, NSAIDs, and gabapentinoids are commonly prescribed nonopioids in postoperative settings. When used in combination, they are more effective in reducing pain and minimizing opioids compared with monotherapy [177,382,383,384]. Around-the-clock oral acetaminophen should be the backbone of postoperative pain regimens because of its safety and low cost, in the absence of acute decompensated liver disease [178,385]. Compared with the oral route, intravenous acetaminophen administration may offer faster onset and better analgesia thirty minutes after administration, but overall drug exposure after repeated doses and general clinical benefits are not significantly different [176,386,387,388]. Additionally, the intravenous formulation may impose financial toxicity without additional benefit in patients with functional gastrointestinal tracts as discussed previously [389,390,391].

**Table 7 healthcare-09-00333-t007:** Nonpharmacologic Interventions for Postoperative Analgesia and Comfort.

Category	Examples
Behavioral/cognitive	Progressive muscle relaxation, mindfulness meditation, art therapy, guided imagery/audio-visual distraction
Psychological	Cognitive behavioral therapy (CBT), acceptance and commitment therapy (ACT), locus of control assessment
Environmental	Music, lighting, comfort items, sleep hygiene (e.g., ear plugs, eye shield), personal hygiene (e.g., shower, hair or nail care)
Physical	Heat, ice/cooling, physical therapy, repositioning, acupuncture, massage, osteopathic manipulation, tai chi, yoga, nutrition counseling, healing touch therapy, reiki
Activities	Hobbies/leisure (e.g., playing cards, magazines/books, puzzles, games, journaling, knitting), relaxation (e.g., stress ball, television), pet visitation
Spiritual	Religious literature & services, onsite spiritual counseling

References: [55,163,347,378,380,392].

Selective COX-2 inhibitors or other NSAIDs should be incorporated into most postoperative pain regimens with consideration of the type of surgery, renal function, and cardiovascular risk factors (see Section 3.2). Since inflammation is a key driver of pain after surgery, early anti-inflammatories may be the most effective postoperative analgesic strategies, as evidenced by their superior performance over opioids in analyses of randomized controlled studies [164,393,394,395,396]. Novel intravenous formulations of ibuprofen and diclofenac currently have limited roles in therapy due to a lack of demonstrated superiority to ketorolac and significantly higher cost [214,215]. Escalating doses of ketorolac greater than 10–15 mg per dose and ibuprofen greater than 400 mg per dose may offer additional analgesic benefit, and the duration of ketorolac therapy should generally be limited to no more than 5 days [212,397,398,399,400]. Gabapentin or pregabalin should be considered for patients with neuropathic pain and may help reduce postoperative opioid use in select patients (see Section 3.2). If initiating postoperative gabapentinoids, dose reductions and close monitoring should be provided for the elderly, those with impaired renal or lung function, and those on multiple narcotic medications [191]. Genetic phenotypes at multiple metabolic enzymes contribute to variation in patient response to NSAID and other nonopioid analgesics, and emerging guidelines provide therapeutic recommendations [184,401].

Other nonopioid agents including cannabinoids, muscle relaxants, and tricyclic antidepressants cannot be recommended for routine postoperative use based on available data but may have roles in select surgical populations (e.g., chronic pain, spinal surgery) [144,217,402,403]. Analyses of the endocannabinoid system suggest certain cannabinoid receptors mediate pain sensitization and hyperalgesia, possibly increasing risk of acute pain conversion to chronic pain. Cannabinoids may therefore be detrimental in the acute pain setting despite being beneficial in chronic pain management [150,153,154,404].

#### 3.5.2. Postoperative Opioid Considerations

In addition to nonopioid analgesia, many patients undergoing major painful procedures may benefit from short-term postoperative opioid therapy. Table 8 provides a comprehensive example of postoperative opioid and nonopioid medication orders. As with nonopioid agents, oral opioids should be used preferentially over intravenous agents for patients who can utilize oral administration. The intravenous route does not confer superior efficacy and carries greater risk for adverse events, and should therefore be reserved for patients unable to use the oral route or patients with severe pain that is refractory to increased doses of oral agents [15,38,405]. When the intravenous route is intermittently warranted for severe breakthrough pain, healthcare provider administration of opioid doses according to patient-reported and functional pain assessments is typically adequate, especially for opioid-naïve inpatients. The sublingual and subcutaneous routes are also reasonable, but the intramuscular route should be avoided due to delayed and erratic absorption [15]. One single-center retrospective cohort study suggests sublingual opioids can be utilized for postoperative breakthrough pain with comparable efficacy as the intravenous route, and the sublingual route was associated with reduced opioid-related respiratory depression [346].

When complete reliance on the intravenous route is considered necessary due to severe gastrointestinal dysfunction or surgical need for strict bowel rest, patient-controlled analgesia (PCA) is recommended over intermittent bolus by healthcare providers by some guidelines [24,403]. This notion is increasingly challenged by enhanced recovery practice, however, especially in minimally invasive colorectal surgery [24,406,407]. Providers may consider reserving use of PCA for patients with acute on chronic pain or otherwise requiring significant amounts of intermittent IV opioids, and only until other routes can be used. Maximizing multimodal therapies in earlier phases of care, especially regional anesthesia or lidocaine infusions, may allow for avoidance of PCA in routine patients undergoing colorectal surgery [24]. The use of intraoperative methadone (see Section 3.3.2) or the sublingual route of administration for postoperative opioids are also promising modalities that could be explored for reducing reliance on PCAs. Medication and patient safety issues abound with PCAs [408,409]. Accordingly, average duration of PCA use has been discussed as a quality indicator of hospital opioid stewardship practices [38]. Use of PCAs should be guided by institutional order sets with pre-built doses stratified for opioid-naïvety and risk for opioid-related respiratory depression, and continuous infusions should generally be avoided in opioid-naïve patients [15,71,408,409].

Empiric opioid selection should align with generally preferred agents, patient-specific pharmacologic needs, and the oral route of administration. Oxycodone, hydrocodone, and hydromorphone should be used preferentially due to their decreased propensities for active metabolites, accumulation in end organ dysfunction, drug-drug interactions, and histamine release (Table 9) [410,411,412,413,414]. Morphine, tramadol, and codeine are significantly metabolized to active metabolites and heavily renally eliminated, increasing the risk of adverse effects in some patient populations [410,415]. Codeine and tramadol have limited roles in postoperative pain management due to well-documented interindividual variability in efficacy and safety [416,417]. Polymorphisms at CYP2D6 and drug-drug interactions significantly affect codeine bioactivation to morphine, the pathway most responsible for analgesic efficacy. Likewise, tramadol is metabolized by CYP2D6 into an active metabolite more potent than the parent drug. Patients possessing increased metabolic variants at CYP2D6 (1.5–9.5% of the worldwide population) are at heightened risk of adverse effects from these agents due to greater conversion to active metabolites, and patients with poor metabolizer phenotypes (25.3–70.3% of the worldwide population) may report decreased efficacy from reduced bioactivation [410,411,412,417,418]. These medications should be avoided in most patients since phenotype testing is not routinely performed before prescribing and since multiple agents with more favorable safety and efficacy profiles exist.

Individual patient response to preferred opioids still varies substantially. Genetic polymorphisms affecting opioid metabolism are not uncommon, so rotation to an agent utilizing an alternative metabolic pathway should be considered in patients with unexplained lack of response and/or significant intolerance (e.g., extreme nausea and vomiting with or without insufficient analgesia from oxycodone may be remedied by change to hydrocodone or hydromorphone) (Table 9) [414,418,419]. Newer opioid agonists can also be considered. Oxymorphone may be advantageous in cases of persistent opioid overexposure related to altered metabolism from phase I enzymatic alterations and/or significant renal impairment. Tapentadol is unique in pharmacologic and pharmacokinetic profiles and can be a valuable option in cases of significant widespread opioid intolerance, but is completely reliant on renal function for excretion. While tramadol is also sometimes considered in patients with intolerance to preferred opioids, its diverse receptor profile confers increased adverse event risks that are especially undesirable in the postoperative period, in addition to previously discussed risks related to its metabolic pathways [417,420,421,422,423,424,425,426,427,428]. Pharmacists can also assess medication regimens for clinically significant drug-drug pharmacokinetic interactions, especially in patients on antiepileptic medications, azole antifungals, or rifampin [413,429,430]. The interprofessional team should also evaluate for pharmacodynamic interactions affecting the patient’s response, such as additive toxicity risk with concomitant sedatives or anticholinergics.

While allergic reactions to opioids are frequently reported, true IgE-mediated hypersensitivity is rare. Only 15% of patients referred for drug provocation testing due to concern with anaphylactic opioid reactions were diagnosed with opioid allergy in one analysis, and opioids are believed to be implicated in less than 2% of all cases of intraoperative anaphylaxis [431,432]. Angioedema and hemodynamic instability are more likely to indicate true hypersensitivity than other reactions [431,433]. In cases of true opioid hypersensitivity, opioids of different structural classes are unlikely to demonstrate cross-allergenicity, though this risk remains uncertain. The majority of opioid reactions are not mediated by IgE but by mast cell degranulation, however, and may present as hives, hypotension, urticaria, pruritus, and/or severe anaphylactoid responses. More synthetic opioids exhibit decreasing rates of opioid-mediated histamine release, so should be considered in cases of pseudoallergy [431,432,433,434].

Clinicians should adjust the empiric postoperative pain management plan in cases for efficacy and tolerability, taking into account the duration and intensity of expected pain for the specific surgical procedure [15]. The use of “may repeat” doses and separate orders only for breakthrough pain can usually allow for a workable escalation pathway for uncontrolled pain within standardized postoperative order sets, as displayed in Table 8. Incomplete analgesic response precluding usual postoperative functional progress despite these orders should prompt a 25–50% increase to the first-line opioid order dose, based on severity of ongoing pain and in the absence of dose-limiting adverse effects. Breakthrough pain regimens should generally be limited to the first 24 postoperative hours, with acceptable pain control maintained by adjusting oral doses if needed. Adjusting opioid regimens in longer-term pain and in cancer-related pain is discussed extensively elsewhere [71,435]. Patients with adequate analgesia but experiencing ORAEs should be assessed for opioid dose reductions, and all opioids should be tapered after surgery as acute postoperative pain improves. If usual surgical recovery is inhibited by unsuccessful functional pain management and/or unacceptable adverse effects despite appropriate multimodal therapies and patient-specific opioid optimization, postoperative pain management specialty consultation is advised. Acute and transitional pain services for surgical patients are evolving, and have been associated with reduced opioid use and length of stay [113,436,437,438,439,440,441].

**Table 9 healthcare-09-00333-t009:** Opioid Properties to Consider When Selecting or Modifying Postoperative Regimens.

Opioid *(Structural Class)*	Major Metabolic Pathways	Active Metabolites	Effects of End Organ Function ^1^
*Phenanthrene opium alkaloids–highest rate of histamine release*			
Morphine, Codeine (after bioactivation) ^2^	UGT2B7 (phase II metabolism)	Extensive production of active metabolites	Renal impairment significantly increases exposure
*Semisynthetic phenanthrene derivatives of opium alkaloids–cross-reactivity possible between agents*			
Oxycodone	CYP3A4 (primary), CYP2D6 (minor)	Produces small amounts of oxymorphone and other active metabolites	Renal impairment mildly increases exposure
Hydrocodone	CYP3A4 (primary), CYP2D6 (minor)	Produces small amount of hydromorphone and other active metabolites	Not significantly altered by renal impairment
Hydromorphone	UGT2B7 (phase II metabolism)	Multiple active metabolites but clinically unimportant	Not significantly altered by renal impairment
Oxymorphone	UGT2B7 (phase II metabolism)	Metabolites have little activity	Not significantly altered by renal impairment
*Synthetic phenylpropylamine derivatives of opioid alkaloids–cross-reactivity with phenanthrenes unlikely*			
Tapentadol	Unspecified glucuronidation	No active metabolites	Renal impairment significantly increases exposure
Tramadol	CYP2D6, CYP3A4	Extensive production of active metabolites by CYP2D6	Renal impairment increases exposure

^1^ All listed opioids should be reduced in cases of significant hepatic impairment. ^2^ Codeine is a prodrug of morphine (activated by CYP2D6) and is not recommended for postoperative pain management; see text. Abbreviations: CYP = cytochrome P450 enzyme superfamily, i.e., hepatic enzymes responsible for phase I metabolism. References: [178,410,411,412,414,415,423,425,426,429,430].

Despite employing opioid minimization and evidence-based opioid selection when treating postoperative pain, the interprofessional team should actively anticipate and mitigate opioid-related adverse events (ORAEs, Table 10). Nausea/vomiting, constipation, pruritus, respiratory depression, sedation, and delirium continue to be common adverse effects negatively affecting postoperative outcomes and costs of care [6,7,8]. Sedation and respiratory depression are the most concerning ORAEs and should be actively mitigated through institutional monitoring protocols based on current practice guidelines and published literature. Protocols should include the use of the Pasero Opioid-Induced Sedation Scale (POSS) and capnography monitoring in addition to conventional respiratory parameters and nursing assessments [15,442,443,444,445,446]. Avoiding concomitant sedatives, especially benzodiazepines, to all feasible extent is also an important modifiable risk for postoperative respiratory depression, sedation, and delirium. This is crucial in patients with higher baseline risks for this complications, including the elderly, obese, and those with preexisting lung disease [38,143,190,447,448,449,450,451,452]. Specialized monitoring for patients receiving perioperative neuraxial opioids must be standardly executed and supported by institutional order sets as outlined elsewhere [15,453]. Some enhanced recovery guidelines recommend against routine intrathecal opioids as this strategy may not have a positive benefit-risk profile in this setting [188].

Patients prescribed opioids should also receive scheduled stimulant bowel regimens to avoid opioid-induced constipation and progression to ileus, a risk that is heightened in the postoperative period (Table 10). Standard preventative use of a stimulant laxative such as senna or bisacodyl is generally effective in preventing opioid-induced constipation in opioid-naive patients, and available evidence does not suggest a superior agent [454,455,456,457,458]. The addition of stool softeners (i.e., docusate) and/or laxatives of alternative classes (e.g., osmotic agents like polyethylene glycol or magnesium oxide) may be added if needed postoperatively, but sugar-based strategies such as lactulose or sorbitol should be avoided due to adverse event risks [454,455]. Unique considerations exist in major colorectal surgery and are discussed in enhanced recovery guidelines [281]. Peripherally acting opioid antagonists have been developed to combat opioid-induced constipation with mixed results for clinical outcomes and cost-effectiveness related to postoperative ileus [459,460,461,462]. Naloxegol and alvimopan may have comparable efficacy in the postoperative period [463]. An alternative agent used in chronic constipation, lubiprostone, does not appear to have superior efficacy over senna in the postoperative setting [464].

#### 3.5.3. Postoperative Considerations in the Opioid-Tolerant and/or Substance Use Disorder Populations

Postoperative pain management in patients with preexisting opioid tolerance and/or substance use disorders is more complicated and high-risk than that of opioid-naïve counterparts, and specialist consultation is strongly advised [15,18,36]. Nonopioid medications and nonpharmacologic options are especially important in this population due to significant opioid receptor up-regulation. In the opioid-tolerant surgical patient, multimodal analgesia may help limit opioid dose escalation, reduce the incidence of adverse events, and facilitate faster postoperative opioid weaning. Stronger consideration should be given to postoperative use of gabapentinoids, ketamine, and regional anesthesia than what may be used in opioid-naïve patients.

Empiric as-needed opioid regimens should be dosed with consideration to baseline opioid use and closely monitored, recognizing that higher doses and/or longer tapers may be warranted. Patients on preoperative opioids have increased risk for suffering if undertreated and increased rates of ORAEs if overexposed. Still, opioids should be utilized only after first-line administration of nonopioids and used at the lowest effective dose, avoiding persistent dose escalations in the postoperative period [18]. To this end, opioid-exposed patients (i.e., those with preoperative opioid use below 60 MED) can usually be prescribed routine postoperative opioid orders as for opioid-naïve patients, with increased monitoring and adjustment for efficacy as needed. Truly opioid-tolerant patients (i.e., those with preoperative opioid use ≥60 MED) should be interviewed to discern their precise preoperative daily utilization to inform a patient-specific postoperative opioid regimen. Postoperative opioids should not be dosed solely upon prescription drug monitoring program (PDMP) data to avoid unnecessary narcotic exposure in patients taking less than maximum quantities prescribed. Opioid-tolerant patients undergoing minor procedures may only warrant routine as-needed opioid dose orders (e.g., oxycodone 5 mg q4h PRN, may repeat within 1 h if ineffective) in addition to their baseline opioid exposure.

After major painful procedures, opioid-tolerant patients often warrant opioid exposure equivalent to a 50–100% increase from their baseline MED to achieve adequate analgesia and functional outcomes in the immediate postoperative period. Some literature suggests postoperative opioid requirements up to four times that of opioid-naïve patients may be necessary after the same procedure, and little published guidance exists on how best to accomplish this [18,117,128]. Chronic opioid requirements may be maintained by modestly increasing the patient’s usual as-needed opioid dose at the same dosing interval, with additional orders as-needed for breakthrough pain. Alternatively, opioid doses could be scheduled throughout daytime hours to provide the patient’s baseline MED, with additional as-needed doses to allow for adequate control of postoperative pain. A third option may be to order the patient’s usual as-needed opioid dose at a shorter dosing interval (e.g., every 3 h as needed instead of every 4 h) with a breakthrough pain option. To illustrate, a patient regularly taking oxycodone 10 mg every 4 h throughout the day prior to admission (i.e., 60–75 MED baseline use) could be ordered one of the following sets of empiric opioid orders upon postoperative inpatient admission after a major painful procedure, assuming the oral route of administration for primary analgesia and the sublingual route for breakthrough pain:(a)oxycodone 10 mg PO q4hr PRN moderate-to-severe pain, may repeat 5 mg dose within 1 h if pain unrelieved; oxycodone 5 mg SL q4hr PRN moderate-to-severe breakthrough pain × 24 h(b)oxycodone 10 mg PO q4hr scheduled while awake; oxycodone 5 mg PO q4hr PRN moderate-to-severe pain; oxycodone 5 mg SL q4hr PRN moderate-to-severe breakthrough pain × 24 h(c)oxycodone 10 mg q3hr PRN moderate-to-severe pain; oxycodone 5 mg SL q4hr PRN moderate-to-severe breakthrough pain × 24 h.

All initial opioid options are in addition to maximal scheduled nonopioid and nonpharmacologic orders, and accompanied by close monitoring for any appropriate adjustments. Orders for opioids as-needed for breakthrough pain should generally still be limited to the immediate postoperative period (i.e., order should automatically expire after the first 24 h of inpatient ward admission). Ongoing need for breakthrough pain opioid doses should prompt evaluation for nonsurgical causes of pain, further optimization nonopioid therapies, and an increase to the primary as-needed opioid order on a patient-specific basis.

Patients with chronic pain and/or opioid use disorders may benefit from a patient-controlled analgesia (PCA) modality when pain is very difficult to control or when the oral route cannot be used [15,117,128,468]. Empiric reliance on intravenous opioids via PCA is increasingly falling out of favor, however, and should not be viewed as routinely necessary in colorectal surgery when enhanced recovery and multimodal analgesia modalities are maximized [24,406]. Experts are increasingly finding this to be true even in opioid-tolerant patients, and opioid-free intraoperative analgesia is even being explored in this population [18]. If PCAs are employed for opioid-tolerant patients, dosing should be patient-specific after assessment of baseline opioid use, as discussed in detail elsewhere [71,117,128,469].

Continuation of chronic long-acting pain medication regimens is recommended, in consultation with the patient’s outpatient prescriber (see Section 3.1.3). Chronic buprenorphine or methadone therapy should be continued either at baseline dosing regimens or by dividing the total daily dose throughout the day to maximize their analgesic activity (see Section 3.1.3). The patient’s usual total daily dose, or a slightly increased total daily dose, is divided into 2 to 4 doses throughout the day starting on the day of surgery. The patient can then be discharged on their usual preoperative regimen without therapy interruption [121,125,128]. Alternatively, some have advocated for a buprenorphine dose reduction in the perioperative period if the patient is on higher chronic doses and/or is experiencing inadequate pain relief despite appropriately dosed as-needed opioids, citing the dose-dependent mu opioid receptor antagonism of buprenorphine [119,122,126,132]. Patients on maintenance buprenorphine or methadone must also be ordered as-needed opioids at tolerant doses (see examples provided earlier in this section) to effectively treat postoperative pain in addition to the continued buprenorphine/methadone regimen, regardless of the dosing strategy employed for them.

Despite available evidence and guidance, healthcare providers may carry prejudices that result in under-treatment of postoperative pain in the opioid-tolerant and/or opioid use disorder populations. Such misconceptions often include that maintenance therapy with buprenorphine or methadone alone provides sufficient postoperative analgesia, that additional opioids for analgesia may cause addiction relapse or undue respiratory depression risk, or that the use of patient-controlled analgesia (PCA) may exacerbate these risks. In actuality, receptor up-regulation and the pharmacology of these agents confer the need for additional short-acting opioids at opioid-tolerant doses in order to provide equipotent analgesia to that provided to opioid-naïve patients. Available evidence does not support that this strategy exacerbates substance use disorders or increases risk for respiratory depression when appropriate dosing and monitoring are employed. Conversely, under-treated pain is likely a more significant risk factor for opioid misuse, ORAEs, and relapse [74,128,470].

### 3.6. Discharge Phase

Discharge opioid prescribing following surgery has significantly contributed to the ongoing U.S. opioid epidemic [29]. Collaborative discussions surrounding discharge opioid prescribing are imperative to minimize the risks of dependency and misuse, and should include all analgesics that are to be continued after discharge. Enhanced recovery programs that integrate standardized opioid-sparing analgesic regimens have significantly reduced or eliminated opioid use in the postoperative setting [13]. Opioid-sparing analgesics should therefore be optimized during the inpatient stay and continued at discharge. Postdischarge multimodal analgesia has been associated with decreased outpatient opioid consumption after major procedures [471]. Duration of opioid-sparing analgesics after hospital discharge should be tailored to the individual needs of the patient and the anticipated length of pain expected after surgery. To mitigate adverse effects and dependence, prescriptions for NSAIDs and gabapentinoids should generally be limited to 1–2 weeks postdischarge. If refills are to be prescribed, an evaluation from a prescriber should be conducted to assess etiology of ongoing pain and appropriateness of continued therapies [472].

Until recently, evidence-based guidelines on postoperative opioid prescribing were not readily available. Variable and often excessive opioid quantities have been prescribed after surgery, especially in the U.S. [4,473]. In 2016, the Michigan Opioid Prescribing Engagement Network (OPEN) released procedure-specific guidelines to help reduce overprescribing of opioids after surgery. These guidelines are adjusted regularly using expert opinion, patient claims data, and evidence-based literature, and are only intended for patients who are considered opioid-naïve [32]. Since implementation at 43 hospitals, there has been a significant reduction in the quantity of opioids prescribed after surgery and a corresponding reduction in opioid consumption by patients [474]. Subsequently, multiple other collaboratives have also published postoperative opioid prescribing guidelines for adults [30,31,475,476] and for children [477].

These guidelines should be used as a foundation to inform procedure-specific institutional practices for opioid prescribing at the point of hospital discharge after surgery. However, opioid prescribing must be individualized within this framework. The patient’s pain control and opioid use in the 12–24 h preceding discharge should be evaluated before prescribing discharge analgesics [478]. Patients undergoing minor procedures, those experiencing minimal pain, or patients who are opioid-naïve may not require opioid prescriptions at discharge. When opioids are prescribed to the opioid-naïve patient population, it is best practice to minimize the duration of supply to three days or less for procedures associated with rapid recovery from severe pain, seven days or less for medium term recovery procedures, and fourteen days or less for expected longer term recovery procedures [31]. Long-acting opioids should not be prescribed for the management of acute postoperative pain after discharge and should be especially avoided in patients who were previously opioid-naïve [15,32]. Opioid-tolerant patients generally have higher opioid requirements than opioid-naïve patients and prescribing a postdischarge opioid taper for this patient population is recommended. Typically, tapering the opioid dose by 20–25% every one to two days is tolerated by most patients as their pain is improving [15]. Detailed postoperative opioid taper examples are presented elsewhere [478]. Additionally, prescription drug monitoring programs (PDMPs) should be reviewed prior to prescribing opioids at discharge to chronic opioid users. This allows for review of the patient’s current home supply and prevents overprescribing of unnecessary opioids at discharge [478].

Despite successful institutional efforts to decrease inpatient opioid prescribing, this has not necessarily translated into reduced opioid quantities prescribed at hospital discharge [479]. Discharge analgesic prescriptions are therefore unlikely to correlate with inpatient orders unless enhanced recovery pathways also have effective transitions of care procedures in place. This should include multidisciplinary communication informing patient-specific prescriptions as opposed to “per protocol” discharge opioid prescriptions for a given procedure. Additionally, data is emerging that shared decision-making, where patients are able to play a role in the amount of opioids they are prescribed at discharge, in conjunction with patient discharge education, can reduce the number of pills prescribed [480]. When considering reduced opioid quantities at discharge, a common concern among surgeons is an increase in office calls from patients requesting opioid prescription refills. Ample evidence supports that a large portion of opioids prescribed at discharge after surgery go unused, however, and initiatives to limit discharge opioid prescription quantities have successfully reduced opioid exposure without adversely affecting pain management or refill requests [42,44,45,46,93,473,476,481,482,483,484,485,486,487,488,489,490]. Maximizing nonopioid therapies and developing patient-specific plans are essential to the success and safety of such practice changes.

Pain management exit plans (PMEP) are an excellent resource for all postoperative patients, especially those with high opioid requirements [478]. Exit plans provide a detailed summary of the analgesics prescribed at discharge, including how each medication should be taken, common side effects, and appropriate disposal techniques (Figure 2). Exit plans focus on multimodal analgesia with an emphasis on nonopioids as the mainstay of therapy. If opioids are prescribed, a taper is developed and outlined in the PMEP using the lowest effective dose. Attention to tablet size and formulation should be considered for those given a taper in order to improve patient compliance. Note that splitting tablets can be challenging for some and use of whole tablets may be preferred for those undergoing a taper. Combination opioid products (e.g., oxycodone/acetaminophen) should be avoided in discharge opioid prescriptions since they limit the ability to safely maximize opioid-sparing analgesia throughout the recovery phase.

Discharge counseling, with an emphasis on nonopioid analgesics as first line therapy, is essential for safe and successful postoperative pain control [15,101,478]. Discharge counseling should be pursued in conjunction with a PMEP or other standardized educational tool and may be completed by a pharmacist, pharmacy or medical student, advanced practice provider, or physician. Patients being discharged with opioid prescriptions should be educated about proper opioid storage and disposal. Opioids should be kept in a locked cabinet, away from children, pets and friends or family. Storing opioids appropriately can reduce accidental overdoses and decrease opioid diversion, since a majority of people who misuse opioids obtain them from a friend or family member [491]. Providers may consider involving a family member to secure and administer the medication to provide accountability and reduce temptation for opioid misuse or diversion in at-risk patients. If able, facilities dispensing opioid prescriptions should provide safe, at-home medication disposal systems to encourage appropriate and prompt disposal of unused opioids [480,492,493,494]. Other disposal methods include medication collecting bins, often found in hospitals, pharmacies, or police stations, and community medication take back events. As a last resort, patients may consider mixing unused medications in a plastic bag with coffee grounds or cat litter and disposing of them in the household trash. Flushing unwanted medications down the toilet should be discouraged as this leads to pharmaceutical contamination of the water supply [27,100,478,495,496].

Careful attention to the quantity of opioids prescribed at discharge to patients planning to resume medical marijuana or other illicit substances, such as heroin, is vital. In 2018, 67,367 drug overdoses were reported in the U.S., with 69.5% involving opioids [497]. Incidence of opioid overdose after postoperative discharge is greatest in the early period, and estimated to be 26.3 events per person-year during the first thirty postoperative days [498]. Co-prescribing of naloxone, a rapid-acting opioid antagonist, should therefore be considered at the point of postoperative discharge for patients at risk of opioid overdose. These patients may include those prescribed more than 50 MED per day, patients prescribed concomitant benzodiazepines, and patients with a history of respiratory disease, substance use disorder, or mental health disorders [54,499,500]. Naloxone may also be prescribed to patients if they are concerned about opioid misuse in their household.

While acute pain management prescribing is the responsibility of the surgical team, collaboration with chronic pain prescribers and/or addiction medicine specialists is crucial for successful postoperative pain control and mitigation of adverse events in these high-risk populations. This communication can help prevent relapse in those with a history of substance use disorder and promote a smooth transition to maintenance medication regimens; hence, the outpatient provider should be engaged before surgery and as soon as feasible after discharge [104,119]. For patients on chronic buprenorphine, therapy should almost always be continued perioperatively, including at the point of hospital discharge, in addition to a short-acting full mu-opioid agonist prescription for acute pain management where usually indicated [119,126,132]. Surgical providers should ensure the patient has enough buprenorphine to last until they can see their buprenorphine prescriber, contacting the prescriber to troubleshoot any foreseeable gaps. Ideally, this appointment should be within 3 days of discharge. As an alternative to the “bridge prescription,” patients can return to the emergency department for administration of buprenorphine for up to 72 h after discharge. For methadone, if the patient’s home dose was decreased or split during the perioperative period, the dose should generally be returned to home dosing at discharge. Arrangements must be made for the patient on methadone to go to their clinic the following day to receive their medication. It is imperative to discontinue chronic naltrexone products at discharge and to defer their reinitiation to the outpatient prescriber after the patient has been off of opioids (see also Section 3.1.3) [117,124].

### 3.7. Follow-Up Phase

Development of persistent opioid use is a risk when prescribing opioids for the treatment of acute pain. This risk is amplified by increased doses, additional days supplied, and duration of use. The likelihood of long-term opioid use significantly increases after five days of opioid therapy [501]. For this reason, patient follow-up should ideally take place within five days of discharge, particularly for those who were prescribed opioids. Follow-up may be conducted in person or via telemedicine. A mobile phone app, downloaded by the patient prior to hospital admission, has been shown to effectively monitor patient pain and opioid requirements after surgery. The patient answers daily mobile phone app questions that include pain assessment. These data are reviewed and pain management revisions are implemented at an in-person or telemedicine clinic visit within 4–7 days after discharge [502].

Follow-up assessments should evaluate ongoing postoperative pain, opioid and nonopioid use, and the status of unused opioids. The pain evaluation should assess pain trajectory, which includes pain intensity as well as time to resolution of pain. Patients identified as having an abnormal pain trajectory (e.g., those experiencing numeric pain scores greater than four on postoperative days three-seven) have been found to have a higher risk of developing persistent postoperative pain and should be monitored closely [503]. Closer follow up may also be warranted in those with a history of substance use disorder or those with mental health comorbidities.

Patients identified as having difficulty with postoperative pain control should receive education about proactive pain management. By taking scheduled doses of nonopioid medications, patients are able to “stay ahead” of their pain and prevent severe pain breakthroughs. For those struggling to wean off of opioids, providers should further optimize nonopioid medications, reiterate nonpharmacologic modalities, and encourage opioid tapers whenever possible. Pain management exit plans can be employed as they are at hospital discharge or updated in the outpatient setting, and should be strongly considered in this patient population [478]. The need for additional opioid prescriptions should be limited and assessed on a case-by-case basis, e.g., in opioid-tolerant patients requiring longer tapers. Coordination with the patient’s other outpatient providers is important, and opioid refills from both surgical and nonsurgical providers should be accounted for [504].

For patients with unused opioids, medication disposal education should be reiterated. Providing patients with local medication take-back locations or safe disposal devices can facilitate appropriate narcotic disposal and limit redistribution within the community [492,493,494].

## 4. Interprofessional Collaboration in Sustaining Perioperative Performance Measures Related to Pain Management and Opioid Prescribing

### 4.1. From the Surgical Institution Perspective

Pain assessment and management metrics have been critical focus areas for healthcare institutions in recent decades, sometimes with deleterious effects. In 2001, as part of a national effort to address the widespread underassessment and undertreatment of pain, The Joint Commission (formerly The Joint Commission on the Accreditation of Healthcare Organizations or JCAHO) introduced pain management standards for healthcare organizations [505]. While well-intended, the standards were also informed by an unfortunately misguided understanding of the addictive potential of opioids at the time [3,506]. This practice movement ultimately resulted in the elevation of pain as the “fifth vital sign”, giving pain equal status with blood pressure, heart rate, respiratory rate, and temperature. Nurses were required to assess pain as an objective sign, instead of as a subjective symptom of surgical recovery [507,508,509]. Hospitals have also been incentivized to improve patient satisfaction with pain management via the Centers for Medicare & Medicaid Services (CMS) Value-Based Purchasing (VBP) program, which adjusts each hospital inpatient payment according to its performance on quality measures [510]. One tool used to evaluate quality measures within VBP is the Hospital Consumer Assessment of Healthcare Providers and Systems (HCAHPS) survey. This survey is administered to patients after hospital discharge and previously asked patients how often hospital providers did, “everything in their power to control your pain” [505]. By directly linking patient satisfaction with pain management to hospital compensation, the survey may have incentivized opioid overprescribing [511].

The Joint Commission and other organizations have since recognized a need to modify standards to mitigate unintended consequences in the wake of the ensuing opioid epidemic [3,359,508,509,512]. The Joint Commission revised their pain standards to include an emphasis on patient safety and the promotion of multimodal analgesia in 2018 [3,36]. Additionally, the revised pain management-related HCAHPS questions shifted from a focus on the perceived quality of pain management efforts to quality of communication about pain management [513]. Furthermore, many U.S. states have enacted opioid prescribing restrictions affecting surgical providers [35]. These revised standards and a shifting paradigm to reduce opioid prescribing are driving surgery centers to reevaluate their approach to perioperative pain management. The requirement by many states to review prescription drug monitoring programs (PDMPs) when prescribing opioids has been linked with a reduced rate of opioid prescriptions in hospitals [514]. Additionally, The Joint Commission requires hospitals to collect and analyze data to monitor their ability to safely prescribe opioids, an important step in the effort to demonstrate reductions in perioperative opioid prescribing without negatively impacting the quality of pain management [67].

In addition to reimbursement-driving quality metrics and legal pressures, healthcare institutions are motivated by increased transparency of their patients’ pain management-related outcomes. Tools such as the CMS Hospital Compare websites and Leapfrog Hospital Safety Grade are available online to consumers [515,516,517]. These quality data are influenced by subjective patient satisfaction indicators in addition to objective outcome metrics. Evaluations of elective surgical programs, such as those providing hip and knee replacements, are therefore only an internet search away from prospective patients. Evidence suggests that an institution’s reputation for postoperative pain management has an important influence on potential healthcare consumers. A recent study assessed the preferences of hip and knee arthroplasty patients regarding publicly available quality metrics. This discrete choice experiment yielded that patients are willing to accept suboptimal hospital ratings and facility cleanliness in exchange for better postoperative pain management and complication rates [518].

Some institutions have implemented opioid stewardship programs (OSPs) to achieve these goals. Core pillars of OSPs include interprofessional collaboration on protocols and services related to multimodal pain management, education on opioid prescribing and stewardship to staff and providers, education to patients, caregivers and community members on safe opioid use and disposal, opioid-related risk reduction, and data analysis and reporting of related quality metrics [38,66,68,519,520,521,522]. An expert panel has proposed quality indicators for measuring opioid stewardship interventions in hospital and emergency settings. These nineteen measures assess quality of inpatient pain management, opioid prescribing practices, ORAE prevention, and transitions of care [38,523].

Although current quality standards and market incentives better align with shared goals by patients, providers, and institutions, the cost of nonopioid medications can pose a barrier for institutions to implement multimodal analgesia throughout perioperative care. Intravenous acetaminophen (pending the widespread availability of this formulation from generic manufacturers in early 2021), intravenous NSAID formulations, and liposomal bupivacaine represent newer nonopioid interventions that drive analgesics to rank among the most expensive therapeutic drug categories [524]. The substantial cost of these agents relative to conventional generic medications may contribute to overreliance on cheap, widely available opioid medications in the perioperative setting [391]. Fortunately, collaborative investigator-initiated research has provided comparative efficacy data to inform cost–benefit comparisons between some of these high-cost agents and their conventional counterparts [176,268,270]. Interprofessional stewardship efforts have demonstrated success in mitigating the potential financial toxicity of perioperative multimodal analgesia by limiting such high-cost agents to populations unable to achieve the same degree of benefit from conventional alternatives [390,525].

It has long been recognized that successful perioperative care involves interdisciplinary collaboration among surgeons, anesthetists, medicine physicians, nurses, and physical therapy providers. Perhaps historically underrecognized has been the value of the clinical pharmacist in improving perioperative patient outcomes and efficiencies [526]. Despite well-supported benefits to diverse patient outcomes and care teams, pharmacists may be underutilized in postoperative pain management. As pharmacotherapy experts with a longitudinal view of the perioperative care continuum, pharmacists are well-poised to perform or oversee many important functions to optimize surgical patient analgesia and institutional opioid stewardship efforts [27,478,527]. These may include completing pre-admission medication reconciliation, advising on preoperative optimization and planning for perioperative management of chronic pain therapies, developing standardized preemptive analgesic protocols with appropriate patient-specific adjustments, supporting intraoperative multimodal analgesic use through protocol development, education, and operationalization, managing postoperative analgesic therapies, advising on discharge opioid and nonopioid prescribing, developing patient educational materials and providing discharge counseling, and assessing patients at follow-up to optimize opioid tapers and screen for postoperative complications [68,478,528,529]. One pre- and post-intervention study spanning 6 years evaluated the impact of a pharmacy-directed pain management service that performed both consult-based and stewardship functions at a large public hospital. The service was associated with decreased total institutional opioid use, increased nonopioid analgesic use, fewer opioid-related respiratory depression events, and ongoing improvement in pain-related HCAHPS patient survey domains [530]. Similarly, a pharmacist-led post-discharge opioid deescalation service was implemented at a major tertiary institution for orthopedic surgery patients recently discharged from the institution’s acute pain service. In the published evaluation of this service, the post-intervention group realized similar pain intensity ratings with significantly lowered opioid doses and incidence of constipation [437]. Healthcare institutions may therefore consider investment in pharmacy services to help drive quality improvement and cost-savings initiatives related to postoperative pain management and opioid stewardship.

### 4.2. From the Surgeon Perspective

The surgeon perspective of best-practices evidence-based perioperative performance is a team approach within standardized enhanced recovery pathways. Each member of the perioperative interdisciplinary team provides valuable knowledge that contributes to opioid stewardship efforts. Where resources are available, perioperative pain management and opioid stewardship is ideally pharmacist-led, from preoperative evaluation through the inpatient stay and postdischarge follow-up [531]. Described below is an example of the teamwork required in a colorectal enhanced recovery pathway to minimize opioid use while effectively treating postoperative pain.

Nonopioid pain management options are optimized throughout the care continuum for all patients on the surgical service. Through preadmission screening, an enhanced recovery nurse navigator may identify patients with a history of chronic opioid use. This allows the pharmacist to contact the patient and develop a focused perioperative pain management plan. Anesthetists are other important enhanced recovery collaborators. Their expertise in perioperative pain management and postoperative nausea and vomiting (PONV) prevention assist with minimizing the need for opioids. Enhanced recovery patients without complications typically receive transversus abdominis plane (TAP) blocks in the preoperative suite from the anesthetist. Postoperative patients are never “nothing by mouth” after surgery when awake and alert, therefore, enhanced recovery postoperative orders should not routinely include intravenous opioids. The pharmacist leads the multimodal pain management strategy at daily inpatient interdisciplinary rounds that include surgeon, resident surgeon, physician assistant, case manager, social worker, enterostomal nursing, and patient care unit nursing staff. Knowledgeable patient care nurses, well-informed in pain management goals and providing consistent care plan messages to patients, are an integral component of standardized perioperative pain control.

Surgeon opioid and nonopioid discharge prescriptions are written in consultation with the enhanced recovery team pharmacist and are based on inpatient pain control and opioid needs in the 12–24 h leading up to discharge. Pain management exit plans are developed by the pharmacist and provided to those with high opioid requirements. Patients receiving an exit plan are seen by pharmacy and educated about the importance of multimodal analgesia and opioid tapers. One study showed that a pharmacist-led enhanced recovery pain management plan resulted in less than 50% of patients requiring opioid prescriptions at the time of discharge for patients having robotic colorectal surgery. The average number of 5 mg oxycodone tablets prescribed in those who received prescriptions was 6 to 8 while the average number used was 2.5 to 3 tablets. Only 0.5% to 0.75% of patients required opioid prescription refills [531].

Perioperative pain management and opioid stewardship continues after patient discharge in the surgeon clinic. One study showed that enhanced recovery pharmacist participation in an early post-discharge clinic where all postoperative patients are seen within 4–7 days of discharge maximized assessment of pain management and reinforcement of nonopioids as the primary pain management option. Additionally, overall readmission rates were significantly decreased, especially with postoperative pain as a readmission diagnosis [502]. In addition to improved patient outcomes, longitudinal involvement of clinical pharmacists in perioperative pain management has been associated with surgical provider satisfaction [528]. Pharmacists may therefore be valuable to optimizing patient care and in maximizing surgeon resources.

Pain management in enhanced recovery is therefore a dynamic, collaborative, interprofessional effort that requires reassessment and evidence-based changes. A prospectively maintained database allows real-time collection and evaluation of enhanced recovery data that includes opioid and nonopioid information [65]. Implementation of an opioid stewardship program is applicable to all surgical specialties and should be incorporated into enhanced recovery pathways.

### 4.3. From the Patient Perspective

Patient-centered outcomes and the surgical patient experience should remain the focus of collaborative care and process improvement. Clinical practice guidelines endorse an individualized approach to all aspects of postoperative pain management based on patient needs and preferences, and echo the need to engage patients in shared decision-making throughout this process [15]. Available evidence suggests clinical pharmacists can positively impact patient experience indicators related to postoperative pain management. The incorporation of clinical pharmacists into patient education prior to joint arthroplasty was associated with modest increases in pain-related domains of the HCAHPS satisfaction survey [532]. A comprehensive clinical pharmacy service in a total joint arthroplasty population at another institution included preoperative education, postoperative pain management optimization, and discharge counseling interventions. This service was associated with improved patient understanding of discharge medications and patients indicated a high degree of satisfaction with pharmacist interactions [529].

To illustrate the importance of postoperative pain management and opioid stewardship to the patient perspective, the following account was authored by a colorectal surgical patient of two of the authors and published with his permission (edited only for brevity):

#### 4.3.1. Preparing for Surgery

“For patients, fully understanding how surgery will affect them physically and emotionally and what type of pain management practices will be employed both before and after is a critical first step if they are to take charge of their own health care. Surgery is a scary proposition for the patient. If you add in the anticipated discomfort and pain it only escalates the unknown, elevating fear and anxiety. Speaking from experience, this quickly takes center stage in a patient’s mind. With my three surgeries, I found it essential to take ownership and control and learn as much as I could about these surgeries and my recovery. Fully understanding possible surgical risks and complications, as well as the overall goal and expected positive outcomes, was vital if I was going to gain mental control of a challenging health situation.

Most patients do not realize the power exists within themselves to take better control of their surgical outcomes. Deciding on my frame of mind and focusing on the positives, rather than the negatives, immediately put me in a better position to reach my recovery goals. As I saw it, I had two choices: (1) I could worry about the possible complications associated with surgery. If I took that route, I was sure to be miserable, anxious, and not fully connected with my end goal; or (2) I could prepare by becoming knowledgeable about my surgery, perceiving it as another life challenge that would enable me to continue living and to improve my quality of life. For me, surgical challenges are a lot like flying a kite. If you run your kite before the wind, you cannot take off and fly. You have to turn into the wind and face it head-on. The challenge you push against is the very force that lifts you. Therefore, it was clear to me I had to face the headwinds.”

#### 4.3.2. The Enhanced Recovery Program, Phone Applications, and Opioid Use

“My three surgeries would involve perioperative pain control, with transverse abdominis plane (TAP) or epidural pain blocks and a combination of oral pain medications including acetaminophen, ibuprofen, gabapentin, and oxycodone. Today, I’m a veteran when it comes to pain medication, but the real inspiration that empowered me and gave me reassurance that I could make a significant contribution to my recovery was the Enhanced Recovery Program (ERP) offered by my health provider. Along with that, I was able to use a phone application when I returned home. This application allowed me to have morning check-ins with my health-care team, if needed. My health provider also offered an informative class several weeks before my surgery that gave me valuable and concise information to help me understand my upcoming procedure and how to prepare for it and my hospital stay. The class also gave me important information about my postoperative care and recovery at home.

The ERP gave me the reassurance and tools needed to control my health care, creating a solid foundation for a good outcome. After attending this class, I realized that I had the power to actively engage as a patient who can contribute, participate, and determine outcomes. I was no longer a bystander but a player in this game. This significantly reduced my anxiety, replacing it with positive energy. I honestly believe this shortened my recovery time for all three surgeries.

In the ERP class, a nurse navigator and a pharmacist addressed the most concerning aspect of my surgery: How to control pain? I realized I am afraid of pain. I honestly believe all of us are. However, getting preoperative education on pain management led to the insight that I needed to take control of my pain rather than let it control me. Understanding opioid risks and benefits gave me the confidence and courage to set a goal to get them out of my life after surgery as soon as reasonably possible. Most of us are keenly aware of the opioid crisis still raging both worldwide and in the United States. I was initially concerned that this might eventually be me.

Well, it could have been me. All of us can be throttled by addictions when we least expect it. However, the underlying key to my success was the preoperative and postoperative education I received. What I did learn, and benefit from was the powerful combination of ibuprofen and acetaminophen and how they work together very well to relieve surgical pain. After stopping opioids, I was continued on a regimen of these over-the-counter pain relievers and quickly discovered my pain was being managed without the use of narcotics. This alternate step was presented and outlined in my ERP class. This was an enabler for me, and I was able to be more mentally alert, have less constipation issues and feel comfortable enough to go home. The ERP umbrella provided an open and honest conversation through clear and straightforward directions about what must be done before and after surgery. ERP and the medical staff gave me realistic and attainable goals for my recovery. I was a partner in my own health-care decisions, and I took ownership for my successful recovery. The well-trained medical staff promptly addressed my concerns. The addition of the phone application, which I found to be an excellent communication tool, provided me much needed emotional reassurance and support before, during, and after surgery.”

#### 4.3.3. Lessons Learned

“As a frequent-flyer patient with lots of surgeries, treatments, and narcotics use, I can report that I landed safely back in my everyday life. Additionally, this was mainly because of the expert care as well as the comprehensive education I received from the medical staff, doctors, pharmacists, and nurses. In all cases, my ERP experience gave me the solid foundation I needed to empower myself and focus on the win, not the illness. I discovered journaling every day with accompanying photos, audio, and video. I now have five solid years of life experience, good and bad, that I can look back on.

All of us will eventually face fragility and mortality. However, for this patient, my medical experiences and the numerous medical staff who helped me during trying times have given me the gift of life. I am grateful that I was forced to confront an often inevitable part of being alive and to now fully understand that we as patients can take ownership of and apply direction to our recoveries.”

## 5. Conclusions and Future Directions

While myriad multimodal strategies exist, ongoing comparative assessments of analgesic combinations and anesthetic approaches within enhanced recovery practice are warranted to further understand and optimize perioperative patient care. Novel analgesic agents and modalities continue to be developed, and their place in therapy should be thoughtfully studied [56,286,533,534,535,536]. Pharmacogenomic assessments show promise in elucidating precision pain management [537,538]. Additional evaluation of the influence of perioperative analgesic strategies on the development of persistent postoperative pain and opioid use would be an invaluable contribution to the literature [2,50,539]. Implementation studies describing successful opioid stewardship programs should be pursued to address practice challenges and increase universal adoption [38,68,540].

Effective perioperative pain management requires a multifaceted team-based approach that begins prior to admission and continues after discharge. Healthcare providers must collaborate throughout institutional practice and process improvement with the shared goals of providing optimal patient care while minimizing opioid exposure. Standardized perioperative pathways should maximize nonpharmacologic therapies and multimodal analgesics, provide decision-support for the judicious use of opioids, and include mitigation strategies for ORAEs and postsurgical opioid dependence. Collaborative practice models should ensure appropriate patient-specific application of available strategies to high-risk and/or opioid-tolerant surgical populations. Pain and addiction medicine specialist consultation, transitional pain services, and opioid stewardship programs should be appropriately resourced across healthcare systems and surgery centers. Incorporating evidence-based pain management and opioid stewardship strategies into a standardized perioperative program will support safe, high-quality, and consistent surgical patient care.

## Figures and Tables

**Figure 1 healthcare-09-00333-f001:**
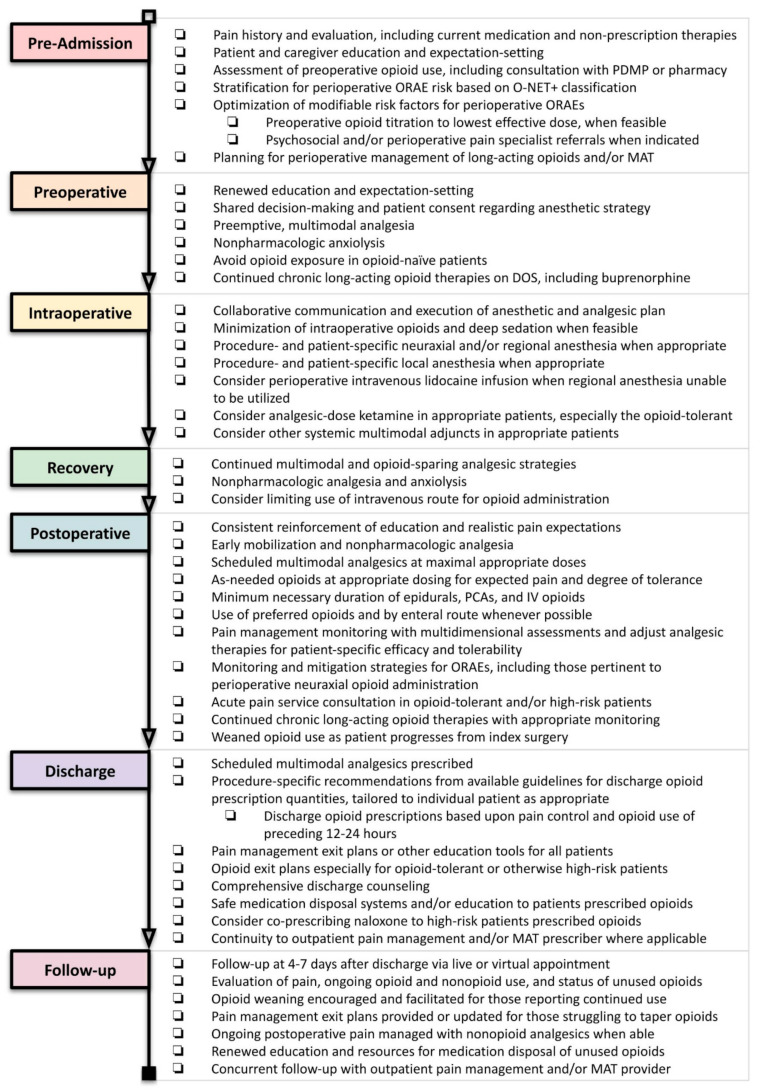
Perioperative Pain Management and Opioid Stewardship Interventions across the Continuum of Care. Legend: DOS = day of surgery, IV = intravenous, MAT = medication-assisted treatment (i.e., for substance use disorders), O-NET+ = opioid-naïve, -exposed or -tolerant, plus modifiers classification system, ORAE = opioid-related adverse event, PCA = patient-controlled (intravenous) analgesia, PDMP = prescription drug monitoring program.

**Figure 2 healthcare-09-00333-f002:**
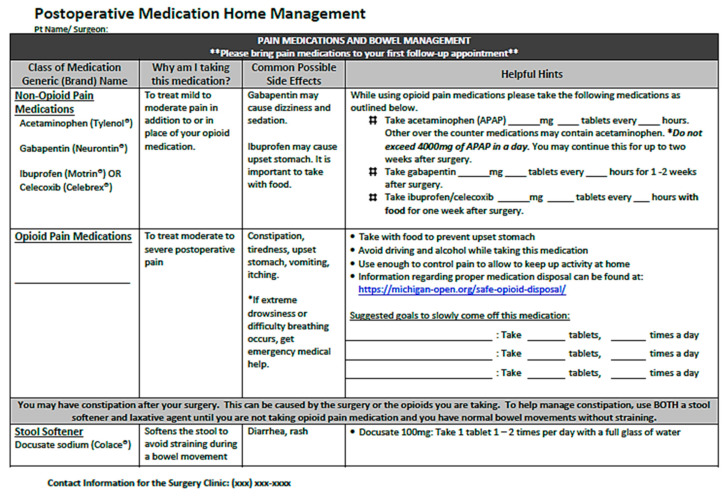
Example of a Pain Management Exit Plan (PMEP) to be used at postoperative hospital Discharge.

**Table 1 healthcare-09-00333-t001:** Current Recommendations for Equianalgesic Dosing of Opioids Commonly Encountered in Perioperative Settings.

Drug	Equianalgesic Doses (mg)	
	IV/IM/SC ^1^ Dose	PO/SL Dose
Oxycodone ^2^	10	20
Hydrocodone ^3^	N/A	25
Hydromorphone ^4^	2	5
Morphine ^3^	10	25
Fentanyl	0.15	N/A
Oxymorphone	1	10
Tapentadol	N/A	100
Tramadol ^2^	100	120

^1^ The IM route of administration is not recommended. ^2^ IV formulation not available in the U.S. at the time of this writing. ^3^ Oral equianalgesic dose equivalent of 30 mg has been used and is also reasonable, given variations in bioavailability between morphine/hydrocodone and oxycodone (equianalgesic ratio ranges from 1:1 to 2:1 morphine:oxycodone based on individual patient absorption). ^4^ Previous resources have used a 1:5 ratio for parenteral:oral hydromorphone, but newer data suggest a ratio 1:2.5 is more appropriate. IM = intramuscular, IV = intravenous, mg = milligrams, N/A = not applicable, PO = oral, SC = subcutaneous, SL = sublingual. Adapted from *Demystifying Opioid Conversion Calculations: A Guide for Effective Dosing, 2nd Edition, 2019* [71].

**Table 2 healthcare-09-00333-t002:** O-NET+ Classification System and Recommended Optimization for Patients on Preoperative Opioids.

Step 1: Classify Preoperative Opioid Exposure and Presence of Risk Modifiers		
Opioid-Naïve	No opioid exposure	In the 90 days prior to DOS
Opioid-Exposed	Any opioid exposure <60 MED	In the 90 days prior to DOS
Opioid-Tolerant	Any opioid exposure ≥60 MED	In the 7 days prior to DOS
+ Modifiers	+ Uncontrolled psychiatric conditions (e.g., depression, anxiety) + Behavioral tendencies likely to impact pain control (e.g., pain catastrophizing, low self-efficacy) + History of SUD (e.g., substance dependency, alcohol or opioid use disorders) + Surgical procedure associated with persistent postop pain (e.g., thoracotomy, spinal fusion)	
**Step 2: Stratify Risk for Perioperative ORAEs**		
Opioid-Naïve	+ No modifiers	→ Low Risk
	+ 1 modifier	→ Moderate Risk
	+ ≥2 modifiers	→ High Risk
Opioid-Exposed	+ No modifiers	→ Moderate Risk
	+ ≥1 modifier(s)	→ High Risk
Opioid-Tolerant	+ No or any modifiers	→ High Risk
**Step 3: Recommend Risk-Stratified Pre-Admission Optimization**		
Low Risk	Preoperative education and perioperative multimodal analgesia	
Moderate Risk	Preoperative education and perioperative multimodal analgesia +	
	Preoperative psychological optimization	
High Risk	Preoperative education and perioperative multimodal analgesia +	
	Preoperative psychological optimization +	
	Preoperative referral to perioperative pain specialist	

Abbreviations: DOS = day of surgery, MED = oral morphine equivalents per day, O-NET+ = opioid-naïve, -exposed, or -tolerant plus modifiers, ORAE = opioid-related adverse event, SUD = substance use disorder. Adapted from [18].

**Table 3 healthcare-09-00333-t003:** Recommendations for Perioperative Management of Long-Acting Opioids and Medication Assisted Therapy (MAT).

Medication	Perioperative Plan ^1^	Postoperative Plan ^1^
**Long-acting pure mu-opioid agonists for chronic pain** (e.g., OxyContin^®^), including continuous transdermal use (e.g., Duragesic^®^) or intrathecal infusions	Continue typical dose throughout periop period including on DOS, in addition to sufficient intraop analgesia	Continue typical dose and provide opioid-tolerant dosing for PRN opioid orders, consider PCA if expect significant pain
**Methadone**	Continue typical dose throughout periop period including on DOS, in addition to sufficient intraop analgesia	Continue typical dose, may divide into q6-8hr dosing to maximize analgesic benefit Provide opioid-tolerant dosing for PRN opioid orders
**Buprenorphine oral, sublingual, and buccal formulations** (e.g., Suboxone^®^, Subutex^®^, Belbuca^®^), including combination products with naloxone	Option 1: Continue typical dose ^2^ throughout periop period including on DOS, in addition to sufficient intraop analgesia	Continue typical dose and provide opioid-tolerant dosing for PRN opioid orders
	Option 2 (*consider if high risk for relapse and/or very painful procedure*): Continue typical dose through day prior to surgery; temporarily increase and/or divide dosing into shorter intervals starting DOS, in addition to sufficient intraop analgesia	Continue increased and/or divided buprenorphine regimen and use opioid-tolerant dosing for PRN opioid orders Discharge on original/typical buprenorphine regimen with sufficient opioid-tolerant PRN opioid supply
**Buprenorphine transdermal patch, subdermal implant, or subcutaneous implant** (e.g., Butrans^®^, Probuphine^®^)	Continue typical dose throughout periop period including on DOS, in addition to sufficient intraop analgesia	Continue typical dose and provide opioid-tolerant dosing for PRN opioid orders
**Naltrexone oral formulations** (e.g., ReVia^®^, Contrave^®^)	Discontinue 3 days prior to surgery and hold on DOS, provide usual intraop analgesia	Continue to hold therapy postop, provide opioid-naïve dosing for PRN opioid orders with close monitoring ^3^ Discontinue naltrexone at discharge and reinitiate with outpatient prescriber after pain recovery complete
**Naltrexone extended-release IM injection** (e.g., Vivitrol^®^)	Ideally schedule surgery for ≥4 weeks after last injection and hold throughout periop period, provide usual intraop analgesia	

^1^ All patients should receive maximal multimodal pharmacologic and nonpharmacologic adjuncts across their care continuum as discussed in other sections, and all changes to chronic therapies should be made in concert with the managing prescriber. ^2^ Some have advocated for preoperative dose reduction in patients on total daily doses ≥12–16 mg; see discussion. ^3^ Patients on chronic naltrexone therapy may exhibit increased sensitivity to opioids after naltrexone discontinuation due to opioid receptor up-regulation; increased monitoring for adverse events is warranted. Abbreviations: DOS = day of surgery, IM = intramuscular, intraop = intraoperative, periop = perioperative, PCA = patient-controlled analgesia, PRN = as needed. References: [18,116,117,119,120,121,122,123,124,125,126,127,128].

**Table 4 healthcare-09-00333-t004:** Example Preemptive Analgesia Protocol.

Drug ^1^	Dose	Exclusions ^2^ and Comments
Acetaminophen	975 mg	Exclude in patients with acute decompensated liver failureDo not exclude in patients with chronic liver disease
Celecoxib ^3^	400 mg if <65 years old, 200 mg if ≥65 years old	Exclude in patients with any current or preexisting renal impairment and in those undergoing cardiac surgery Do not exclude due to sulfa allergies
Gabapentin	300 mg if <65 years old, 100–300 mg if ≥65 years old or if any renal impairment	May consider avoiding in patients at high risk of respiratory depression, delirium, or dizziness, if risks outweigh opioid-sparing benefits

^1^ All to be given as one-time medication orders by mouth in preoperative holding area within 2 h of incision, unless exclusion is met. ^2^ These in addition to patients with true significant allergy to drug. ^3^ Additionally, reduce dose by 25–50% if known CYP2C9 poor metabolizer. References: [15,60,165,166,168,170,180,181,182,183,184].

**Table 6 healthcare-09-00333-t006:** Clinical Considerations for Intraoperative Systemic Multimodal Analgesics.

Drug [refs]	Dosing ^1^	Potential Benefits	Monitoring and Cautions ^2^
**Lidocaine** [15,18,26,33,57,261,288,289,290,291,292,301,302,303,304,305,306,307]	0.5–1.5 mg/kg loading dose over 10 min then 1–1.5 mg/kg/h infusion through end of procedure Infusions continued or instated postop at 0.5–1 mg/min in some protocols with appropriate monitoring, though some recommend limiting to ≤24 h Always dose based on IBW and do not exceed max exposure of 120 mg/hr	Provides improved pain control, decreased opioid use May decrease risk of persistent postop pain, increase functional recovery, decrease ORAEs, and hasten bowel recovery May decrease cancer recurrence, though further study is needed	Avoid in patients with significant end organ dysfunction, certain cardiac abnormalities ^3^, uncontrolled seizure disorders, electrolyte imbalances, during pregnancy, and in those weighing <40 kg Unsafe to combine with most local anesthetic-based regional anesthesia techniques or topical patches (see discussion) Monitoring protocols for cardiac function and LAST prevention
**Ketamine** [15,18,25,33,217,261,308,309,310]	0.1–0.35 mg/kg bolus followed by intraop infusion at 0.1–1 mg/kg/h, and/or postop infusion at 0.1–0.5 mg/kg/h Alternatively, consider 5–10 mg boluses q1hr prn	May decrease risk of persistent postop pain and hasten recovery times Improved pain control and decreased opioid useEvidence of benefits in opioid-tolerant patients Can be given intranasally	Avoid in patients with severe or uncontrolled psychiatric, cardiovascular, or hepatic disease, and in pregnancy Avoid in acute hypertension or tachyarrhythmia and in decompensated patients with high shock index
**Magnesium** [33,297,298,309,311,312,313,314]	1–3 g loading dose over 15 min then 0.5–1 g/h during procedure	May improve antinociception and reduce sedative and opioid requirements similarly to ketamine	Important to monitor BP, HR, RR, and muscle relaxation Caution or avoid in renal insufficiency, neuromuscular disorders, electrolyte imbalances, bradyarrhythmias, hypotension or at high risk for hemodynamic compromise
**Dexmed-etomidine** [33,250,261,315,316,317,318,319,320,321,322]	0.3–1 MCG/kg/h, with or without 0.5–0.6 MCG/kg loading dose over 10 min	May improve pain control, decrease opioid requirements, decrease delirium risk, and inhibit catecholamine surges to mitigate surgical stress and end organ damage, but data is limited	Dose- and rate-dependent bradycardia and hypotension: monitor and titrate carefully or avoid if susceptible May be comparable to IV when added to perineural or neuraxial injections instead, but safety unclear
**Esmolol** [323,324,325]	500 MCG/kg bolus followed by 5–50 MCG/kg/min infusion	May reduce postop pain scores, opioid use, and ORAEs, but evidence is currently limited	Patient selection and monitoring related to systemic beta blocker therapy should apply, including consideration of concomitant beta blocker/AV-nodal blocking therapies
**Dexamethasone** [33,250,254,259,309,326,327,328,329,330,331,332,333]	1–10 mg once at beginning of procedure	May prolong duration of regional anesthesia, reduce pain and opioid use	Systemic corticosteroid administration can contribute to postop hyperglycemia and demargination; comparable efficacy between IV and perineural administration
**Methadone** [334,335,336,337,338,339,340]	0.1–0.3 mg/kg (max 30 mg) once at beginning of procedure	May have additional analgesic benefits similar to ketamine or neuropathic agents May be preferable to high-dose fentanyl or preemptive opioids	Duration of plasma half-life can exceed 24 h—monitor for ORAEs Caution in patients at risk for ventricular dysrhythmias given QTc-prolonging risk

^1^ All agents given intravenously. ^2^ These in addition to patients with true significant allergy to drug. ^3^ Includes second or third degree sinoatrial, atrioventricular, or intraventricular heart block without a functioning artificial pacemaker, Adam-Stokes syndrome, Wolff-Parkinson-White syndrome, or other active dysrhythmia, severe cardiac failure (ejection failure <20%), or concomitant Class I antiarrhythmic. Abbreviations: AV = atrioventricular, BP = blood pressure, HR = heart rate, IBW = ideal body weight, ICP = intracranial pressure, IOP = intraocular pressure, LAST = local anesthetic systemic toxicity, MCG = microgram, mg = milligram, ORAE-opioid-related adverse event, RR = respiratory rate.

**Table 8 healthcare-09-00333-t008:** Example of Postoperative Inpatient Pain Management Orders.

Medication (Route ^1^)	Application	Dose Range ^2^	Comments
**Acetaminophen (PO)**	All patients without contraindication	650 mg PO q4h while awake or 975 mg PO q6h^2^	Selective use of the IV & PR routes may be appropriate, see discussion
**Anti-inflammatory**—Choose one in all patients without contraindication (see Section 3.2)			
Celecoxib (PO)		100–200 mg PO q12–24h^2^	May be preferred to ibuprofen
Ketorolac (IV)		15 mg IV q6h × 24h, max duration 5 days^2^	Limit use to first 24–48 h, change to alternative when can take PO
Ibuprofen (PO)		400 mg PO TID with meals or q6h^2^	
**Neuropathic Agent**—Choose one in patients with significant pain or high opioid use, weighing patient-specific risks and benefits (see Section 3.2)			
Gabapentin (PO)		100 mg PO TID, or 100 mg with breakfast and lunch plus 300 mg qHS dose^2^	Opioid-sparing benefits must be weighed against patient-specific risks for sedation, respiratory depression, and dizziness
Pregabalin (PO)		25–50 mg PO BID^2^	
**Oral As-needed Opioid**—Choose one in patients undergoing painful procedures for duration of expected moderate-to-severe surgical pain, gradually decreasing dose during recovery period			
Oxycodone (PO)		Opioid-naïve: 5 mg PO q4 h PRN moderate-to-severe pain, may repeat 5 mg dose within 1 hr if ineffective (total available range 5–10 mg q4h PRN)	Initial dosing for opioid-tolerant patients should be based upon baseline opioid use, usually allowing for 25–100% increase from baseline exposure in immediate postop period ^4^
Hydrocodone (PO)		Dosing as above, recognizing this is slightly lower analgesic potency (see Table 1)	Decrease or discontinue scheduled acetaminophen to avoid overexposure if using combination products
**As-needed Opioid for Breakthrough pain**—Choose one for first 24 h postop; if used frequently and/or needed beyond immediate recovery phase then assess for other causes of pain and/or increase primary as-needed opioid			
Oxycodone (SL)		5 mg PO/SL q4 h PRN moderate-to-severe breakthrough pain	Consider “may repeat” dose and/or initial 10 mg dose for breakthrough pain in opioid-tolerant patients ^4^
Hydromorphone (IV)		0.2–0.5 mg IV/SC q3 h PRN moderate-to-severe breakthrough pain ^3^	Only order IV opioids for severe breakthrough pain or absolute contraindications to oral analgesia Consider “may repeat” dose and/or initial 0.8–1 mg dose for breakthrough pain in opioid-tolerant patients
**NMDA Antagonist**—Consider in severely painful procedures, in opioid-tolerant patients, or in cases of pain-sedation mismatch in appropriate patients			
Ketamine (IV)		0.1–0.35 mg/kg or 5–10 mg IVP once or q2 h PRN for refractory pain, or in cases of pain-sedation mismatch precluding opioid use	Continuous infusion of 0.05–0.35 mg/kg/hr may be considered postoperatively where supported by institutional protocol

^1^ All represented oral formulations are short-acting/immediate release dosage forms. ^2^ For medications with dosing ranges provided, consider using lower doses within the suggested range for patients with advanced age and/or chronic kidney and liver disease. Patients with chronic pain and and/or opioid use disorders may benefit from higher doses. ^3^ Available concentrations of hydromorphone injectable should determine the measurable dose, within this range, in order to ensure practical drug administration (e.g., rounded doses to the nearest 0.1 mL or 0.25 mL). ^4^ A number of practical strategies exist to accomplish this—see Section 3.5.3). Abbreviations: IV = intravenous, IVP = intravenous push, PO = oral or by mouth, SC = subcutaneous, SL = sublingual.

**Table 10 healthcare-09-00333-t010:** Recommended Monitoring and Mitigation Strategies for Postoperative ORAEs.

ORAE	Monitoring and Mitigation Strategies
Sedation, Respiratory, Depression, Delirium	Vigilant monitoring of respiratory and mental status by validated scales (e.g., POSS) and respiratory function data, especially EtCO2, per standardized institutional protocols based on available guidelines Evaluate for opioid dose reduction and/or rotation Avoid concomitant sedatives, especially benzodiazepines Standard opioid antagonist protocols for urgent/emergent reversal Optimize physical and environmental contributing factors (e.g., allow sunlight in room during daytime hours, limit interruptions to sleep)
Constipation, Ileus	Early ambulation, diet advancement as tolerated, and goal-directed hydration as per surgery-specific enhanced recovery protocol Standard postoperative scheduled bowel regimen started on DOS continued for duration of opioid therapy, including stimulant laxative and stool softener (e.g., senna-docusate 8.6–100 mg PO BID), reduced as opioid requirements decrease and bowel function returns to normal Standard additional PRN laxative for constipation (e.g., polyethylene glycol 17 g daily PRN), escalation to PR suppository in refractory cases
Nausea, Vomiting	Standard postoperative PRN antiemetic orders (e.g., ondansetron 4 mg PO q6hr PRN or droperidol 1.25 mg IV q6h PRN nausea/vomiting) Assess for opioid reduction and/or rotation (see text) Optimize physical and environmental contributing factors (e.g., nutrition, noxious stimuli)
Urinary Retention	Monitor per standard institutional protocol Decrease anticholinergic burden (e.g., remove scopolamine patches, avoid antihistamines) Hold chronic anticholinergic therapies in the immediate postoperative period where possible (e.g., oxybutynin) Avoid neuraxial opioids, consider avoiding neuraxial anesthesia entirely in patients at high risk (e.g., older males with prostate disease)
Pruritus	Low-dose nalbuphine PRN is likely most efficacious and safe strategy and may be warranted for duration of neuraxial opioids in some cases May consider age-appropriate, low-dose antihistamines where needed (e.g., diphenhydramine 12.5–25 mg PO q6hr PRN), but this is less efficacious than nalbuphine and may increase risk for other ORAEs Avoid neuraxial opioids in susceptible patients

Abbreviations: BID = twice daily; DOS = day of surgery; EtCO2 = end-tidal carbon dioxide; ORAE = opioid-related adverse drug event; PO = by mouth/oral; POSS = Pasero Opioid-Induced Sedation Scale, PR = per rectum. References: [15,442,443,444,453,454,455,456,465,466,467].

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
