# Peer review of "Perioperative Pain Management and Opioid Stewardship: A Practical Guide"

_healthcare, 2021, doi:10.3390/healthcare9030333_

Round 1

Reviewer 1 Report

This is a very comprehensive and detailed review of analgesia management strategies. The authors provided a great deal of practically useful details on prescribing practices for reducing the risks of opioid therapy for surgery patients.

However, the Reviews is way too long - it can make a few book chapters, so it requires a significant trimming to improve the impact and make it more legible.

Refer to the role of pharmacogenetics in opioid therapy and pain management and CPIC guidelines for opioids in earlier sections; refer to CPIC Guidelines for TCAs and NSAIDs in relevant sections. 

Specific comments:

Please cut section #2 Statistics and Definitions to one page, and add subheadings. There is a lot of redundancy in general statements and reinforcing sentences. Add subheadings.

Table 2: consider adding benzodiazepines and gabapentinoids as concomitant drugs that can increase the risk of complications and opioid abuse. Clean up colour formatting on fill vs text. (either solid colour throughout or no colour) 

Table 3:

Left adjust text, and add margins on left and right side.

Consider combining naltrexone oral and IV recommendations to reduce word count.

Consider adding recommendations on pre-operative use of antipsychotics, Benzodiazepines and cannabinoids that are known to affect opioid dose requirements and risk of complications with opioid use or refer to other sections and tables.

Table 4 

Celecoxib - recommended dose ranges should be adjusted to the patient's CYP2C9 metabolic status.

Gabapentin - was also shown to increase the risk of complications when combined with opioids (PMID: 33372975), though authors discount the risks in the second last paragraph on page 14.

Table 5 - fix formatting in the first column; add spacing on left and right sides for all columns 

Section 3.3.1. Regional and Local Anesthesia.

Add comments on the risk of cardiovascular complications for patients with Brugada syndrome and Peripheral Regional Anesthesia and Intraoperative Systemic Multimodal Analgesia  https://www.brugadadrugs.org/pref_avoid/ . 

Postoperative Opioid Considerations - Please refer to the Oxford League Table of Analgesic efficacy http://www.bandolier.org.uk/booth/painpag/Acutrev/Analgesics/lftab.html 

Table 8 - fix colour highlights; left adjust; add spacing. 

Author Response

This is a very comprehensive and detailed review of analgesia management strategies. The authors provided a great deal of practically useful details on prescribing practices for reducing the risks of opioid therapy for surgery patients.

-Thank you for your review

However, the Reviews is way too long - it can make a few book chapters, so it requires a significant trimming to improve the impact and make it more legible.

-Have tried to shave down unimportant words in tables and have addressed per specific comments as detailed below

Refer to the role of pharmacogenetics in opioid therapy and pain management and CPIC guidelines for opioids in earlier sections; refer to CPIC Guidelines for TCAs and NSAIDs in relevant sections. 

-Thank you for these references. They have been incorporated into page 22 (nonopioid analgesics) and page 25 (opioids) and applicable table a suggested below

Specific comments:

Please cut section #2 Statistics and Definitions to one page, and add subheadings. There is a lot of redundancy in general statements and reinforcing sentences. Add subheadings.

-The Intro and Stats/Definitions sections have been cut down substantially as suggested to avoid redundancy and subheadings have be added across pages 1-4

Table 2: consider adding benzodiazepines and gabapentinoids as concomitant drugs that can increase the risk of complications and opioid abuse. Clean up colour formatting on fill vs text. (either solid colour throughout or no colour) 

-While we clinically agree with this suggestion, the Table 2 is a representation of the specific risk stratification scheme recommended in the referenced guidelines, hence we do not feel like we can change them here. We do discuss the risk with concomitant benzos elsewhere in the section. Formatting has been fixed as suggested (pg 8). 

Table 3:

Left adjust text, and add margins on left and right side. Formatting adjusted as suggested (pg 10)

Consider combining naltrexone oral and IV recommendations to reduce word count. Combined final column for both as text was the same - effectively reduced word count and table size to more effectively fit on page (pg10)

Consider adding recommendations on pre-operative use of antipsychotics, Benzodiazepines and cannabinoids that are known to affect opioid dose requirements and risk of complications with opioid use or refer to other sections and tables. We felt these recs were beyond the scope of this table as it was titled Long-Acting Opioids and MAT but agree these are important recs and we discuss them on pg 11-12

Table 4 

Celecoxib - recommended dose ranges should be adjusted to the patient's CYP2C9 metabolic status. This rec has been added to table on pg 13 thank you

Gabapentin - was also shown to increase the risk of complications when combined with opioids (PMID: 33372975), though authors discount the risks in the second last paragraph on page 14. Yes this concern is addressed in the Exclusions/Comments section of the table for gabapentin on pg 13 and discussed extensively in the text including this reference on pg 14

Table 5 - fix formatting in the first column; add spacing on left and right sides for all columns Formatting changes complete as suggested and agree table looks much cleaning and consolidated now

Section 3.3.1. Regional and Local Anesthesia.

Add comments on the risk of cardiovascular complications for patients with Brugada syndrome and Peripheral Regional Anesthesia and Intraoperative Systemic Multimodal Analgesia  https://www.brugadadrugs.org/pref_avoid/ . Comment and Reference has been added on page 17 thank you

Postoperative Opioid Considerations - Please refer to the Oxford League Table of Analgesic efficacy http://www.bandolier.org.uk/booth/painpag/Acutrev/Analgesics/lftab.html 

-Thank you this has been incorporated on pg 23

Table 8 - fix colour highlights; left adjust; add spacing. Formatting changes made on page 25 as recommended

Reviewer 2 Report

This review was comprehensive and well-written. I think it will fill a much needed space in the literature by bringing together this information in a practical guide for diverse practitioners in the perioperative setting. The tables and figures provide a succinct look at the most important points related to the topic. The inclusion of a patient perspective is unique in this type of article and serves to further support what I think is your main objective: to provide safe and effective pain management for patients in the perioperative setting. The entire manuscript is well-supported by references to the medical literature. I enjoyed reading this manuscript immensely!

I have only a few minor questions/suggestions:

Page 11, First Full Paragraph - Do you have a reference for the alternative option proposed in the second half of the paragraph?

Page 18, Table 6 - In a few instances in the "Potential Benefits" column, you mention "improved pain" - I think you mean either improved pain scores or pain control.

Page 19, Table 6, footnotes - why is MCG capitalized?

Page 20, Third Paragraph - "μ-opioid" (with the symbol) - in every other instance throughout the paper, you have "mu-opioid" - I suggest you keep it consistent throughout. 

Page 23, Last Paragraph - "multimodal modalities" - I suggest rephrasing as this is confusing/redundant. In other places throughout the manuscript, you have used "multimodal techniques" "multimodal strategies" and "multimodal analgesia" - I think any of these would be fine.

Page 38, Conclusion, First Paragraph - "multimodal analgesic modalities" - Same suggestion as page 23.

Author Response

This review was comprehensive and well-written. I think it will fill a much needed space in the literature by bringing together this information in a practical guide for diverse practitioners in the perioperative setting. The tables and figures provide a succinct look at the most important points related to the topic. The inclusion of a patient perspective is unique in this type of article and serves to further support what I think is your main objective: to provide safe and effective pain management for patients in the perioperative setting. The entire manuscript is well-supported by references to the medical literature. I enjoyed reading this manuscript immensely!

thank you for your review!

I have only a few minor questions/suggestions:

Page 11, First Full Paragraph - Do you have a reference for the alternative option proposed in the second half of the paragraph? Thank you we have added reference to the end of this paragraph now on page 10

Page 18, Table 6 - In a few instances in the "Potential Benefits" column, you mention "improved pain" - I think you mean either improved pain scores or pain control. Yes thank you we have corrected in table 6 on pg 19

Page 19, Table 6, footnotes - why is MCG capitalized? This is recommended tall-man lettering per ISMP to help differentiate between MCG and mg from a safety perspective

Page 20, Third Paragraph - "μ-opioid" (with the symbol) - in every other instance throughout the paper, you have "mu-opioid" - I suggest you keep it consistent throughout. Yes thank you we have aligned to "mu" throughout

Page 23, Last Paragraph - "multimodal modalities" - I suggest rephrasing as this is confusing/redundant. In other places throughout the manuscript, you have used "multimodal techniques" "multimodal strategies" and "multimodal analgesia" - I think any of these would be fine. Thank you we have made the minor refinement here

Page 38, Conclusion, First Paragraph - "multimodal analgesic modalities" - Same suggestion as page 23. Thank you we have made the minor refinement here as well